# PARL: A Unified Framework for Policy Alignment in Reinforcement Learning from Human Feedback

**Souradip Chakraborty**[1]**, Amrit Singh Bedi**[2]**, Alec Koppel**[3]**, Huazheng Wang**[4]**,
Dinesh Manocha**[1]**, Mengdi Wang**[5]**, Furong Huang**[1]

[1]University of Maryland, College Park, [2]University of Central Florida, [3]J.P. Morgan AI Research,
[4]Oregon State University,s [5]Princeton University
`{schakra3}@umd.edu`

## Abstract

We present a novel unified bilevel optimization-based framework, PARL, formulated to address the recently highlighted critical issue of policy alignment in reinforcement learning using utility or preference-based feedback. We identify a major gap within current algorithmic designs for solving policy alignment due to a lack of precise characterization of the dependence of the alignment objective on the data generated by policy trajectories. This shortfall contributes to the sub-optimal performance observed in contemporary algorithms. Our framework addressed these concerns by explicitly parameterizing the distribution of the upper alignment objective (reward design) by the lower optimal variable (optimal policy for the designed reward). Interestingly, from an optimization perspective, our formulation leads to a new class of stochastic bilevel problems where the stochasticity at the upper objective depends upon the lower-level variable. True to our best knowledge, this work presents the first formulation of the RLHF as a bilevel optimization problem which generalizes the existing RLHF formulations and addresses the existing distribution shift issues in RLHF formulations. To demonstrate the efficacy of our formulation in resolving alignment issues in RL, we devised an algorithm named A-PARL to solve PARL problem, establishing sample complexity bounds of order $\mathcal{O}(1/T)$. Our empirical results substantiate that the proposed PARL can address the alignment concerns in RL by showing significant improvements (up to 63% in terms of required samples) for policy alignment in large-scale environments of the Deepmind control suite and Meta world tasks.

## 1 Introduction

The increasing complexity and widespread use of artificial agents highlight the critical need to ensure that their behavior aligns well (AI alignment) with the broader utilities such as human preferences, social welfare, and economic impacts (Frye & Feige, 2019; Butlin, 2021; Liu et al., 2022). In this work, we study the alignment problem in the context of reinforcement learning (RL), because a policy trained on misspecified reward functions could lead to catastrophic failures (Ngo et al., 2022; Casper et al., 2023). For instance, an RL agent trained for autonomous driving could focus on reaching its destination as quickly as possible, neglecting safety constraints (misalignment). This could lead to catastrophic failures, such as causing high-speed accidents. The question of alignment in RL may be decomposed into two parts: *(i) How can we efficiently align the behavior (policy) of RL agents with the broader utilities or preferences? (ii) How to reliably evaluate if the current RL policy is well aligned or not?* Addressing the first question is crucial because it serves as a preventive measure, ensuring that the RL agent operates within desired boundaries from the outset. The second question is equally vital as it acts as a diagnostic tool, enabling real-time or retrospective assessment of the RL agent's behavior which could help identify early signs of misalignment to avoid severe consequences.

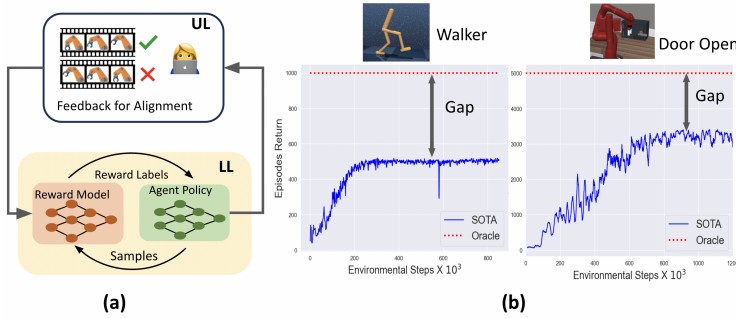

Figure 1: **(a)** This figure shows the proposed PARL framework for policy alignment in reinforcement learning. The standard RL is at the lower level (LL), and the alignment objective is at the upper level (UL). **(b)** This figure shows the performance gap of the SOTA approach due to policy misalignment. The blue curve should be as close as possible to the red dotted line of oracle.

In this work, we propose a novel unified bilevel framework called PARL, capturing the challenge of policy alignment in reinforcement learning using utility or human preference-based feedback. Our PARL framework shown in Figure 1(a) consists of *upper level (UL)* and *lower level (LL)*. The lower level performs the policy alignment step for a given parametrized reward function (addressing question (i)), and the upper level evaluates the optimal lower-level policy to check possible alignment (addressing the question (ii)). A bilevel approach is crucial because of the dependence of the alignment objective on the data generated by the optimal policy of the RL agent.

There are existing approaches such as PEBBLE (Lee et al., 2021) and SURF (Park et al., 2022), which have proposed a heuristic iterative procedure for possible policy alignment in RL but do not focus on the entanglement between the alignment objective and the trajectory collecting policy. This gap (our formulation resolved this) in existing state-of-the-art (SOTA) approaches results in a misaligned objective for alignment evaluation and sub-optimal performance in terms of alignment. We highlight the performance gap of the SOTA approach in Figure 1(b). This gap exists because, without considering the correct evaluation objective at the upper level, existing alignment methods set the initial trajectory for the agent, but they do not offer ongoing assurance of the agent's behavior. Policies can drift or become misaligned due to various factors such as data distribution shifts, model updates, or environmental changes (which we observe in 1(b)). Our proposed formulation provides a rigorous mathematical formulation to address questions (i) and (ii) and improves the performance to achieve policy alignment in practice. We summarize our contributions as follows.

- **Bilevel formulation for policy alignment in RL:** we formulate the RL agent policy alignment as a bilevel optimization problem where the upper-level centers on reward design through policy evaluation over the horizon, and the lower level pertains to policy alignment with the designed reward via policy optimization.

- **Correct evaluation objective for policy alignment in RL:** We highlight the dependence of data distribution of the alignment objective on the RL agent policy, which is missing from the prior research. This provides a reliable objective to evaluate the performance of the current policy of the RL agent at the upper level.

- **Analysis of new class of stochastic bilevel problems:** Interestingly, our bilevel problem does not fall under the class of standard stochastic bilevel optimization problems in the literature (Chen et al., 2022; Kwon et al., 2023; Li et al., 2022b; Ji et al., 2021; Cao et al., 2023; Akhtar et al., 2022; Ghadimi & Wang, 2018b) with the only exception being (Lu, 2023), which also studies coupled decision dependent bilevel optimization problem but not in the context of RL. The unique feature of our problem is to explicitly consider the dependence of data collection at the upper level on the optimal policy parameter at the lower level. We also propose a novel A-PARL algorithm to solve the bilevel problem and derive precise sample complexity bounds of $\mathcal{O}(1/T)$, where $T$ is the number of episodes.

- **Empricial evidence of improved policy alignment.** We evaluate the proposed approach on various large-scale continuous control robotics environments in DeepMind control suite (Tassa et al., 2018) and MetaWorld (Yu et al., 2021). We show that our approach archives better policy alignment due to the use of the corrected upper-level objective we propose in this work. We achieve up to $63\%$ improvement in samples required to solve the task.

## 2 RELATED WORKS

**Preference Based RL.** Revealed preferences is a concept in economics that states that humans in many cases do not know what they prefer until they are exposed to examples via comparisons (Caplin & Dean, 2015). This idea gained traction as it provided a substantive basis for querying humans and elucidating feedback on decision-making problems with metrics whose quantification is not obvious, such as trustworthiness or fairness. Efforts to incorporate pairwise comparison into RL, especially as a mechanism to incorporate preference information, have been extensively studied in recent years – see Wirth et al. (2017) for a survey. A non-exhaustive list of works along these lines is (Roth et al., 2016; Hill et al., 2021; Wirth et al., 2017; Zhu et al., 2023; Xu et al., 2020; Saha et al., 2023; Chakraborty et al., 2023c; 2024).

**Inverse RL and Reward Design.** Reward design has also been studied through an alternative lens in which an agent is provided with demonstrations or trajectories and seeks to fit a reward function. Inverse reinforcement learning (Ziebart et al., 2008a; Brown et al., 2019; Arora & Doshi, 2020) learns a reward to learn behaviors deemed appropriate by an expert. On the other hand, imitation learning directly seeks to mimic the behavior of demonstrations (Ho & Ermon, 2016; Kang et al., 2018; Ghasemipour et al., 2019; Xiao et al., 2019). However, acquiring high-quality demonstrations can be expensive or infeasible (Bai et al., 2022; Chen et al., 2023; Wolf et al., 2023). It also sidesteps some questions about whether a reward can be well-posed for a given collection of trajectories. A further detailed context of related works has been discussed in the Appendix B

## 3 PROBLEM FORMULATION

Consider the Markov Decision Process (MDP) tuple $\mathcal{M} := \{\mathcal{S}, \mathcal{A}, \gamma, P, r\}$ with state space $\mathcal{S}$, action space $\mathcal{A}$, transition dynamics $P$, discount factor $\gamma \in (0, 1)$, and reward $r : \mathcal{S} \times \mathcal{A} \to \mathbb{R}$. Starting from state $s \in \mathcal{S}$, an agent takes action $a$, and transitions to $s' \sim P(\cdot \mid s, a)$. The agent follows a stochastic policy that maps states to distributions over actions $\pi_\theta : \mathcal{S} \to \triangle_{|\mathcal{A}|}$, which is parameterized by $\theta \in \mathbb{R}^d$. The standard finite horizon policy optimization problem is

$$\max_\theta V_s(\theta) := \mathbb{E}\left[\sum_{h=0}^{H-1} \gamma^h r(s_h, a_h) \mid a_h \sim \pi_\theta(a_h|s_h), s_0 = s\right], \tag{1}$$

where the expectation is with respect to the stochasticity in the policy $\pi_\theta$ and the transition dynamics $P$. We note that the formulation in (1) learns a policy for a specific reward function $r$ (fixed a priori). As detailed in the introduction, if the reward function $r$ does not capture the intended behavior required by humans, it would lead to learning a misaligned policy. Therefore, policy learning with a fixed reward does not allow one to tether the training process to an external objective such as human preferences (Christiano et al., 2017), social welfare (Balcan et al., 2014), or market stability (Buehler et al., 2019). To address this issue, we propose a bilevel framework for policy alignment next.

### 3.1 POLICY ALIGNMENT IN REINFORCEMENT LEARNING: A BILEVEL FORMULATION

To achieve policy alignment in RL, we consider the following bilevel optimization problem:

$$\text{(upper)} \qquad \max_\nu \quad G(\nu, \theta^*(\nu)) \tag{2}$$

$$\text{(lower)} \qquad \text{s.t.} \ \ \theta^*(\nu) := \arg\max_\theta \mathbb{E}\left[\sum_{h=0}^{H_\ell-1} \gamma^h r_\nu(s_h, a_h) \mid a_h \sim \pi_\theta(a_h|s_h).s_0 = s\right],$$

where $\theta$ is the policy parameter as in (1) and $\nu \in \mathbb{R}^n$ is the reward parameter. We discuss the formulation in (2) in detail as follows.

**Lower Level (LL):** This is the policy learning stage for a given reward parameterization $\nu$. For a fixed $\nu$, LL problem is the same as in (1) but note the explicit dependence of LL objective in (2) on the reward parameter $\nu$ which is different from (1). In this work, we restrict focus to the case that the optimizer $\theta^*(\nu)$ at the lower level is unique, which mandates that one parameterize the policy in a tabular (Agarwal et al., 2020; Bhandari & Russo, 2021) or softmax fashion (Mei et al., 2020a); otherwise, at most one can hope for with a policy gradient iteration is to obtain approximate local extrema (Zhang et al., 2020). We defer a more technical discussion of this aspect to Section 5.

**Upper Level (UL)):** This level (we also call it the designer level) evaluates the optimal policy learned at the lower level and tests for alignment. In (2), we aim to maximize an objective function $G(\nu, \theta^*(\nu))$, which depends on the reward parameters $\nu$ and optimal policy parameter $\theta^*(\nu)$, which is an implicit function of $\nu$. To be specific, we consider a utility (which is the alignment objective) at the upper level of the form

$$G(\nu, \theta^*(\nu)) = \Upsilon(\pi_{\theta^*(\nu)}) + Z(\nu), \tag{3}$$

which is comprised of two terms: a quantifier $\Upsilon(\pi_{\theta^*(\nu)})$ of the merit of design parameters $\nu \in \mathbb{R}^n$, and $Z(\nu)$ representing a regularizer or prior directly defined over the parameters of the reward function. More specifically, the explicit mathematical form of $\Upsilon(\pi_{\theta^*(\nu)})$ decides the quality of policy $\pi_{\theta^*(\nu)}$ by collecting trajectories (denoted by $\tau$) and associating a designer's evaluation reward $U_\nu(\tau)$ given by

$$\Upsilon(\pi_{\theta^*(\nu)}) = \mathbb{E}_{\rho(\tau;\theta^*(\nu))}[U_\nu(\tau)] = \sum_\tau U_\nu(\tau) \cdot \rho(\tau;\theta^*(\nu)), \tag{4}$$

where $\rho(\tau;\theta^*(\nu))$ denotes the probability distribution over the trajectories $\tau$ and can be explicitly expressed as $\rho(\tau;\theta^*(\nu)) = \rho(s_0) \prod_{h=0}^{H_u} P(s_{h+1} \mid s_h, a_h) \pi_{\theta^*(\nu)}(a_h|s_h)$, where $H_u$ denotes the length of upper-level trajectory, $\rho(s_0)$ represent the initial state distribution. We discuss the explicit form of such an objective $U_\nu$ in detail in Section 3.2 and also in Appendix C.

**Remark 1** (**Contrast with Standard Stochastic Bilevel Optimization**). We highlight an important difference between our formulation in (2) and the standard stochastic bilevel optimization (SBO) in literature (Ghadimi & Wang, 2018a; Akhtar et al., 2022). We note that in (2), the distribution of stochastic upper-level alignment objective depends on the lower-level optimal variable (policy in our case). This differs from the existing works SBO where the expectation is over data whose distribution does not depend upon the lower level optimization variable. Hence, developing solutions for (2) exhibits unique technical challenges not found in the prior research.

### 3.2 Reinforcement Learning from Human Feedback (RLHF) - A special case

In this section, we demonstrate how our formulation proposed in (3) generalizes the RLHF paradigm (Christiano et al., 2017; Lee et al., 2021; Park et al., 2022). In RLHF, as proposed in Christiano et al. (2017), operates mainly in three iterative steps: we start by (step 1) learning a policy (say $\pi_{\theta^*(\nu)}$) for a given reward $r_\nu$ by solving $\arg\max_\theta \mathbb{E}\left[\sum_{h=0}^{H_\ell-1} \gamma^h r_\nu(s_h, a_h)\right]$, (step 2) collect human feedback after collecting trajectories (denoted by dataset $\mathcal{D}$) from $\pi_{\theta^*(\nu)}$, (step 3) learn a new aligned reward model $\nu$ by solving $\min_\nu \mathbb{E}_{y,\tau_0,\tau_1 \sim \mathcal{D}}[\ell(\nu; y, \tau_0, \tau_1)]$ where $\ell$ denotes the alignment objective, and then go back to (step 1). This iterative process is repeated over multiple iterations as detailed in Christiano et al. (2017); Lee et al. (2021); Park et al. (2022). Irrespective of its effectiveness in practice, a rigorous mathematical formulation for RLHF is missing from the literature. Furthermore, the existing RLHF pipeline, in step 3, completely ignores the fact that the data $\mathcal{D}$ is the function of $\pi_{\theta^*(\nu)}$, which results in an incorrect objective to learn a reward parameter. This results in policy misalignment and a performance gap, as highlighted in Figure 1(b). We reformulate the RLHF problem following our bilevel formulation as follows

$$\max_\nu \ \mathbb{E}_{y,\tau_0,\tau_1 \sim \rho_h(\tau;\theta^*(\nu))}[y \log P_\nu(\tau_0 > \tau_1) + (1-y) \log P_\nu(\tau_0 < \tau_1)] \tag{5}$$

$$\text{s.t. } \theta^*(\nu) := \arg\max_\theta \mathbb{E}\left[\sum_{h=0}^{H_\ell-1} \gamma^h r_\nu(s_h, a_h) \mid, s_0 = s\right],$$

where $P_\nu(\tau_0 > \tau_1)$ is the probability of preferring $\tau_0$ over $\tau_1$ (denoted by $y = 1$). We can model $P_\nu$ using the well-known Bradley Terry model (Bradley & Terry, 1952) by

$$P_\nu(\tau_0 > \tau_1) = \frac{\exp \sum_{h=0}^{H_u-1} r_\nu(s_h^0, a_h^0)}{\exp \sum_{h=0}^{H_u-1} r_\nu(s_h^0, a_h^0) + \exp \sum_{h=0}^{H_u-1} r_\nu(s_h^1, a_h^1)}, \tag{6}$$

where $(s_h^i, a_h^i)$ denotes state-action pair at $h^{\text{th}}$ time-step in the $i^{\text{th}}$ trajectory. At the upper level in (5), we highlight the dependence of data distribution $\rho_h(\tau; \theta^*(\nu))$ on the parameter $\nu$ via optimal policy $\pi_{\theta^*(\nu)}$. The sampling distribution is given as $\rho_h(y, \tau_0, \tau_1; \theta^*(\nu)) =$

$h(y|\tau_0, \tau_1)P(\tau_0; \theta^*(\nu))P(\tau_1; \theta^*(\nu))$, where, $h(y|\tau_0, \tau_1)$ represents the human distribution of trajectory preference (unknown and realized only through samples). The trajectories $\tau_0, \tau_1$ are sampled as $\tau \sim \rho(\tau; \theta^*(\nu))$ with the policy given by $\pi_{\theta^*(\nu)}(a|s)$. Hence, looking at (5), we note that it matches exactly with our formulation in (2) with upper level objective

$$G(\nu, \theta^*(\nu)) := \mathbb{E}_{y, \tau_0, \tau_1 \sim \rho_h(\tau; \theta^*(\nu))}[y \log P_\nu(\tau_0 > \tau_1) + (1 - y) \log P_\nu(\tau_0 < \tau_1)]. \tag{7}$$

## 4 Proposed Approach: An Algorithm to Solve PARL

As mentioned in Remark 1, the resulting bilevel optimization problem in (2) differs from the standard bilevel problem studied well in the optimization literature. Hence, we cannot directly apply any off-the-shelf algorithm to solve 2. In this section, we provide a step-by-step development of an iterative procedure to solve the policy alignment in (2) based on the bilevel algorithm[1] available in Ghadimi & Wang (2018b). Let us begin by deriving the gradient of the upper objective (cf. (3) and (4)) with respect to design parameter $\nu$, which is given by

$$\nabla_\nu G(\nu, \theta^*(\nu)) = \nabla_\nu \sum_\tau U_\nu(\tau) \cdot \rho(\tau; \theta^*(\nu)) + \nabla_\nu Z(\nu) \tag{8}$$

$$= \sum_\tau U_\nu(\tau) \cdot \nabla_\nu \log(\rho(\tau; \theta^*(\nu))) \cdot \rho(\tau; \theta^*(\nu)) + \mathbb{E}_{\rho(\tau; \theta^*(\nu))}[\nabla_\nu U_\nu(\tau)] + \nabla_\nu Z(\nu)$$

$$= \mathbb{E}_{\rho(\tau; \theta^*(\nu))}[U_\nu(\tau) \cdot \nabla_\nu \log(P(\tau; \theta^*(\nu)))] + \mathbb{E}_{\rho(\tau; \theta^*(\nu))}[\nabla_\nu U_\nu(\tau)] + \nabla_\nu Z(\nu)$$

$$= \mathbb{E}_{\rho(\tau; \theta^*(\nu))}\left[U_\nu(\tau) \cdot \sum_{h=0}^{H_u-1} \nabla_\nu \log \pi_{\theta^*(\nu)}(a_h|s_h)\right] + \mathbb{E}_{\rho(\tau; \theta^*(\nu))}[\nabla_\nu U_\nu(\tau)] + \nabla_\nu Z(\nu),$$

where we have used the log-trick and standard rule of expectation to get the final expression in (8) (Williams, 1992; Sutton et al., 1999).

**Novel terms in gradient evaluation:** We emphasize two terms (a) the *score function* term $\nabla_\nu \log \pi_{\theta^*(\nu)}(a|s)$ in (8), denotes the gradient of logarithm of the optimal policy with respect to the design parameter $\nu$; and (b) the expectation is with respect to the trajectory distribution $\rho(\tau; \theta^*(\nu))$ generated under policy at the lower-level $\pi_{\theta^*(\nu)}$. This term captures the change of optimal policy with respect to the reward parameters. This is crucial because the designer (such as a regulatory body or central planner) at the upper level can directly control the policy learning by modifying the reward parameters. We remark that both these terms are missing from the existing RLHF formulation in literature (such as in the objective in step 1 detailed in Section 3.2).

**Challenges:** However, the estimation of $\nabla_\nu \log \pi_{\theta^*(\nu)}(a_h|s_h)$ is nontrivial as it depends on the solution of the lower-level problem in (2), and therefore requires the evaluation of hypergradient $\nabla_\nu \theta^*(\nu)$. To see that, let us employ the shorthand notation $f_h(\theta^*(\nu)) := \log \pi_{\theta^*(\nu)}(a_h|s_h)$, we can rewrite the gradient[2] as

$$\nabla_\nu f_h(\theta^*(\nu)) = \nabla_\nu \theta^*(\nu) \nabla_\theta f_h(\theta^*(\nu)). \tag{9}$$

Now, from the first order optimality condition for the lower level in (2), it holds that

$$\nabla_\theta V_s(\nu, \theta^*(\nu)) = 0, \tag{10}$$

which is the gradient of lower-level objective with respect to parameter $\theta$ evaluated at the optimal $\theta^*(\nu)$. Now, differentiating again with respect to $\nu$ on both sides of (34), we obtain

$$\nabla_{\nu,\theta}^2 V_s(\nu, \theta^*(\nu)) + \nabla_\nu \theta^*(\nu) \nabla_\theta^2 V_s(\nu, \theta^*(\nu)) = 0. \tag{11}$$

The above expression would imply that we can write the final expression for the gradient in (9) as

$$\nabla_\nu f_h(\theta^*(\nu)) = -\nabla_{v,\theta}^2 V_s(\nu, \theta^*(\nu)) \nabla_\theta^2 V_s(\nu, \theta^*(\nu))^{-1} \nabla_\theta f_h(\theta^*(\nu)). \tag{12}$$

---

[1] We remark that it is possible to follow more advanced bilevel optimization algorithms in the literature to solve the bilevel problem in (2), we resort to the basic algorithm to highlight the novel aspects of our problem formulation for RLHF.

[2] Throughout the analysis, we follow the convention as follows : let's say $\nu \in R^{d_1}$, $\theta \in R^{d_2}$. Hence, gradient terms $\nabla_\nu f_h(\theta^*(\nu)) \in R^{d_1}$, $\nabla_\theta f_h(\theta) \in R^{d_2}$ and hessian term $\nabla_\theta^2 f_h(\theta) \in R^{d_2 \times d_2}$ and Jacobian $\nabla_\nu \theta^*(\nu) \in R^{d_1 \times d_2}$ and subsequently $\nabla_\nu \theta^*(\nu) \in R^{d_1 \times d_2}$.

---

**Algorithm 1** Algorithm for Policy Alignment in Reinforcement Learning (A-PARL)

1: **Input**: Reward parametrization $\nu_0$ policy initialization $\theta^0$, upper and lower-level step sizes $\alpha_u > 0, \alpha_\ell > 0$ respectively f
2: **for all** $t = 0, 1, 2, ..., T - 1$ **do**
3:    **for all** $k = 0, 2, ..., K - 1$ **do**
4:       Sample $N$ trajectories $\tau \sim \rho(\tau; \theta^K(\nu_t))$ and estimate policy gradient $\nabla_\theta V_s(\nu_t, \theta^K(\nu_t))$ from equation (15)
5:       Update the policy gradient parameter as :

$$\pi_{\theta^{k+1}(\nu_t)} \leftarrow \pi_{\theta^k(\nu_t)} + \alpha_\ell \nabla_\theta V_s(\nu_t, \theta^k(\nu_t)) \quad \cdots \texttt{\color{blue}policy update}$$

6:    Update the reward parameterization in the upper-level from equation (14) as :

$$r_{\nu_{t+1}} \leftarrow r_{\nu_t} - \alpha_u \widetilde{\nabla}_\nu G(\nu_t, \theta^K(\nu_t)) \quad \cdots \texttt{\color{blue}reward update}$$

7: **Output:** $\nu_T, \theta^K(\nu_T)$

---

We substitute (12) into (8) to write the final expression for the gradient of the upper objective in (2) as

$$\nabla_\nu G[\nu, \theta^*(\nu)] = \mathbb{E}_{\rho(\tau; \theta^*(\nu))} \left[ U_\nu(\tau) \cdot \sum_{h=0}^{H_u-1} \left[ -\nabla_{v,\theta}^2 V_s(\nu, \theta^*(\nu)) \nabla_\theta^2 V_s(\nu, \theta^*(\nu))^{-1} \nabla_\theta f_h(\theta^*(\nu)) \right] \right]$$
$$+ \mathbb{E}_{\rho(\tau; \theta^*(\nu))} [\nabla_\nu U_\nu(\tau)] + \nabla_\nu Z(\nu). \tag{13}$$

Even for the gradient expression in (13), there are three intertwined technical challenges such as the requirement to estimate $\pi_{\theta^*(\nu)}$, evaluating Jacobians and Hessians of the lower-level problem, and sampling trajectories in an unbiased manner from $\rho(\tau; \theta^*(\nu))$ which depends upon $\pi_{\theta^*(\nu)}$. We write the explicit values of the terms as follows. This establishes a precise connection to the generalized alignment objective.

**Upper-level gradient estimation:** To estimate the gradient of the upper-level objective in (8), we require information about $\pi_{\theta^*(\nu)}$ which is not available in general unless the lower-level objective has a closed-form solution. Hence, we approximate $\pi_{\theta^*(\nu_t)}$ with $\pi_{\theta^K(\nu_t)}$ i.e., running K-step policy gradient steps at the lower level to obtain the approximate gradient of the upper level at $\nu_t$ as

$$\widetilde{\nabla}_\nu G(\nu_t, \theta^K(\nu_t)) = \mathbb{E}_{\rho(\tau; \theta^K(\nu_t))} \left[ U_\nu(\tau) \cdot \sum_{t=0}^{H_u-1} [\widetilde{M}^K(\nu_t) \nabla_\theta f_h(\theta^K(\nu_t))] + \nabla_\nu U_\nu(\tau) \right] + \nabla_\nu Z(\nu_t), \tag{14}$$

where $\widetilde{M}^K(\nu_t) = -\nabla_{v,\theta}^2 V_s(\nu_t, \theta^K(\nu_t)) \nabla_\theta^2 V_s(\nu_t, \theta^K(\nu_t))^{-1}$.

**Lower-level objective gradient, Jacobian, and Hessian estimation:** Here, we derive the gradients for the lower-level objective and, subsequently the Hessian and mixed Hessian terms for our algorithmic description. First, we write down the gradient of lower-level objectives using the policy gradient theorem (Williams, 1992; Sutton et al., 1999) as

$$\nabla_\theta V_s(\nu_t, \theta^K(\nu_t)) = \mathbb{E}_{\rho(\tau; \theta^K(\nu_t))} \left[ \sum_{h=0}^{H_\ell-1} \gamma^h r_{\nu_t}(s_h, a_h) \left( \sum_{j=0}^{h} \nabla_\theta \log \pi_{\theta^K(\nu_t)}(a_j|s_j) \right) \right], \tag{15}$$

where $H_\ell$ is the horizon or the episode length. Similarly, the Hessian of the lower objective is:

$$\nabla_\theta^2 V_s(\nu_t, \theta^K(\nu_t)) = \mathbb{E}_{\rho(\tau; \theta^K(\nu_t))} \left[ \sum_{h=0}^{H_\ell-1} \gamma^h r_{\nu_t}(s_h, a_h) \left( \sum_{j=0}^{h} \nabla_\theta^2 \log \pi_{\theta^K(\nu_t)}(a_j|s_j) \right) \right], \tag{16}$$

Finally, we can write the mixed second-order Jacobian matrix as

$$\nabla_{\nu,\theta}^2 V_s(\nu_t, \theta^K(\nu_t)) = \mathbb{E}_{\rho(\tau; \theta^K(\nu_t))} \left[ \sum_{h=0}^{H_\ell-1} \gamma^h \nabla_\nu r_{\nu_t}(s_h, a_h) \left( \sum_{j=0}^{h} [\nabla_\theta \log \pi_{\theta^K(\nu_t)}(a_j|s_j)]^T \right) \right]. \tag{17}$$

Now, utilizing the expressions in (14)-(17), we summarize the proposed steps in Algorithm 1. We explain the execution of Algorithm 1 with the help of Figure 4 in the Appendix B. We denote the number of lower-level iterates by $K$ for better exposition which is a function of upper iterations $t \in T$ in the convergence analysis. Before shifting to analyzing the convergence behavior of Algorithm 1, we close with a remark.

**Remark 2.** We have only presented the analytical forms of the first and second-order information required to obtain a numerical solver for problem in (2). However, in practice, these update directions are unavailable due to their dependence on distributions $\rho(\tau; \theta^*(\nu))$ and MDP transition model $P$. Therefore, only sampled estimates of the expressions in (14)-(17) are available. For this work, we sidestep this challenge by assuming access to the oracles to (14)-(17). This helps us to focus more on the policy and reward entanglement in our bilevel formulation. We can extend the analysis to stochastic settings utilizing the standard techniques in stochastic optimization literature (Chen et al., 2022; Kwon et al., 2023; Li et al., 2022b; Ji et al., 2021; Ghadimi & Wang, 2018b).

**Remark 3** (**Gradient derivations for RLHF problem in Section 3.2**)**.** For the special case of RLHF discussed in, we provide a detailed derivations for the gradients at the lower and upper level objectives in Appendix D.

## 5 CONVERGENCE ANALYSIS

In this section, we analyze the convergence behavior of Algorithm 1. Since the upper level $G(\nu, \theta^*(\nu))$ is non-convex with respect to $\nu$, we consider $\nabla_\nu G(\nu, \theta^*(\nu))$ as our convergence criteria and show its convergence to a first-order stationary point, as well as the convergence of $\theta^K(\nu)$ to $\theta^*(\nu)$. Taken together, these constitute a local KKT point (Boyd & Vandenberghe, 2004)[Ch. 5] Without loss of generality, our convergence analysis is for the minimization (upper and lower level objectives). We proceed then by introducing some technical conditions required for our main results.

**Assumption 1** (Lipschitz gradient of upper objective)**.** *For any $\nu \in \mathbb{R}^n$, the gradient of the upper objective is Lipschitz continuous w.r.t to second argument with parameter $L_g$, i.e., we may write*

$$\|\nabla_\nu G(\nu, \theta) - \nabla_\nu G(\nu, \theta')\| \le L_g \|\theta - \theta'\|. \tag{18}$$

**Assumption 2.** *For all $s \in \mathcal{S}$ and $a \in \mathcal{A}$, reward function is bounded as $r_\nu(s, a) \le R$ and Lipschitz w.r.t to $\nu$, i.e., $|r_{\nu_1}(s, a) - r_{\nu_2}(s, a)| \le L_r \|\nu_1 - \nu_2\|$.*

**Assumption 3.** *The policy $\pi_\theta$ is Lipschitz with respect to parameter $\theta$, which implies $\|\pi_{\theta_1}(\cdot|s) - \pi_{\theta_2}(\cdot|s)\| \le L_\pi \|\theta_1 - \theta_2\|$ for all $\theta_1, \theta_2$. The score function $\nabla_\theta \log \pi_\theta(a|s)$ is bounded $\|\nabla_\theta \log \pi_\theta(a|s)\| \le B$ and Lipschitz, which implies*

$$\|\nabla_\theta \log \pi_{\theta_1}(\cdot|s) - \nabla_\theta \log \pi_{\theta_2}(\cdot|s)\| \le L_1 \|\theta_1 - \theta_2\|. . \tag{19}$$

*Further, the policy parameterization induces a score function whose Hessian is Lipschitz as*

$$\|\nabla_\theta^2 \log \pi_{\theta_1}(\cdot|s) - \nabla_\theta^2 \log \pi_{\theta_2}(\cdot|s)\| \le L_2 \|\theta_1 - \theta_2\| \text{ for all } \theta_1, \theta_2 . \tag{20}$$

**Assumption 4.** *The value function $V_s(\nu, \theta)$ satisfies the Polyak-Lojasiewicz (PL) condition (with unique minima) with respect to $\theta$ with parameter $\mu$. We denote $\{\lambda(\nabla_\theta^2 V_s(\nu, \theta))_j\}_{j=1}^d$ as the eigenvalues of Hessian matrix $\nabla_\theta^2 V_s(\nu, \theta)$. Although, $V_s(\nu, \theta)$ is non-convex in $\theta$, but follows the restriction on the eigenvalues as $\lambda(\nabla_\theta^2 V_s(\nu, \theta)) \in [-\hat{l}, -\hat{\mu}] \cup [\hat{\mu}, \hat{l}]$.*

Assumption 1 is standard in the analysis of non-convex optimization, and Assumption 2 is common in RL which bounds the reward function value (Zhang et al., 2020; Agarwal et al., 2020; Bhandari & Russo, 2021). In Assumption 3, the score function Lipschitz part is considered in the literature. But because we are dealing with a bilevel formulation here, which requires dealing with evaluating the lower level policy at the upper level, and also utilizing second-order information of the value function (cf. (13)), we need to assume policy and Hessian of policy are Lipschitz as well. Also, the second-order smoothness condition was studied in establishing conditions under which a local extremum is attainable in the non-convex setting (Zhang et al., 2020)[Sec. 5]. Additionally, we need Assumption 4 because we are in a bilevel regime without lower level strong convexity (Huang, 2023; Liu et al., 2023). However, the value function satisfies the Polyak-Lojasiewicz (PL) condition under common policy parameterizations (Bhandari & Russo, 2021; Mei et al., 2020b). $\hat{\mu}, \hat{l}$ defined in Appendix 4. we provide a detailed discuss in Appendix I. Next, we introduce key technical lemmas

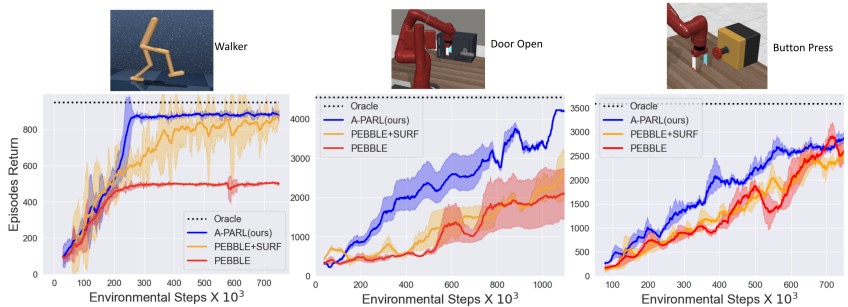

Figure 2: In this figure, we compare the performance of our algorithm A-PARL against SOTA baselines Pebble (Lee et al., 2021), PEBBLE+SURF (Park et al., 2022) and Oracle (true reward) for Walker (DMSuite (Tassa et al., 2018)), DoorOpen and ButtonPress (MetaWorld Yu et al. (2021)) w.r.t ground truth return (averaged over 5 seeds). It clearly demonstrates the superiority of our algorithm over existing baselines in terms of episodic return, where A-PARL achieves near-oracle performance in a much faster time. This highlights the importance of our bilevel framework which considers the dependence (missing from existing literature) of distribution on the lower-level policy parameter during training.

regarding the iterates generated by Algorithm 1. We begin by quantifying the distributional drift associated with the transition model under approximate policy $\pi_{\theta^K(\nu_t)}$ as compared with $\pi_{\theta^*(\nu_t)}$ at the lower level, which results in a transient effect at the upper level.

**Lemma 1.** *Under Assumptions 1 - 4, for trajectory $\tau = \{s_h, a_h\}_{h=1}^{H_u}$, it holds that*

$$D_f(\rho(\tau; \theta^*(\nu)), \rho(\tau; \theta^K(\nu))) \leq \frac{H_u L_2}{2} \|\theta^*(\nu) - \theta^K(\nu)\|, \tag{21}$$

*where $D_f$ is the f-divergence between distributions and $L_2$ is the Lipschitz parameter (cf. Assum. 3).*

The proof of Lemma 1 in provided in Appendix G.1. We highlight that the upper bound in Lemma 1 is novel to our analysis of bilevel RL which will not appear in the RLHF setup considered in the literature. Next, we establish some error bound conditions on key second-order terms that appear in equations (14)-(17) when we substitute $\theta^*(\nu_t)$ by $\theta^K(\nu_t)$.

**Lemma 2.** *Under Assumptions 1-(4), the lower level iterates of Algorithm 1 satisfies $\|\theta^K(\nu_t) - \theta^*(\nu)\|^2 \leq \frac{\eta^K L_6}{\mu} Z$, where, $Z := \max_\nu \|\theta^0 - \theta^*(\nu)\|^2$, $\eta := 1 - \alpha_3$, $\alpha_3 = \alpha_\ell(1 - \frac{\alpha_\ell L_6}{2})\frac{\mu}{2}$, $L_6 = L_5\frac{H L_2}{2} + L_5$, $L_5 = H_l^2 R L_1$, $\mu$ is the PL constant, and $K$ denotes the number of lower-level iterations, and policy gradient step-size satisfies $\alpha_\ell < 2/\min(H_\ell L_2, \mu)$, with $\mu$ as the PL constant (Assumption 4).*

The proof of Lemma 2 is provided in Appendix H.7. The proof relies on the assumption that the value function satisfies a Polyak Lojasiewicz (PL) condition under appropriate policy parametrization.

**Theorem 1.** *Under Assumptions 1-4, for the proposed Algorithm 1, it holds that*

$$\frac{1}{T}\sum_{t=1}^{T}\|\nabla_\nu G(\nu_t, \theta^*(\nu_t))\|^2 \leq \frac{G_0 - G^*}{\delta_1 T} + \frac{\eta \delta_2 L_6 Z}{T \delta_1 \mu(1 - \eta)} \tag{22}$$

*where $G_0 := G(\nu_0, \theta^*(\nu_0))$ and $G^*$ denotes the global optimum of the upper objective, $\delta_1 = \alpha_u\left(1 - \frac{1}{2c_1} - L_g\alpha_u\right)$ and $\delta_2 = \alpha_u\left(\frac{c_1}{2} + L_g\alpha_u\right)$, $c_1$ is a positive constant defined in eqn. (45), and the step-size range of satisfies $\alpha_u < 1/L_g$, with $L_g$ as in Assumption 1 and $\alpha_\ell$ as stated in Lemma 2.*

In Theorem 1, we note that we achieve a final rate of $\mathcal{O}(1/T)$, which matches with bilevel optimization for non-convex upper objective without requiring strong convexity at the lower level in (Ghadimi & Wang, 2018a).

## 6 EXPERIMENTAL EVALUATIONS

We consider the challenging tasks of robotic locomotion and manipulation in the human preference-based RL setting as also considered in Lee et al. (2021); Park et al. (2022); Metcalf et al. (2022).

**Human Feedback:** Although real-human preferences would have been ideal for the experiments, but it is hard to get them. Hence, we leverage simulated human teachers as used in prior research (Lee et al., 2021; Park et al., 2022). The simulated teacher provides preferences on pairwise trajectories to the agent according to the underlying true reward function, which helps us to evaluate the policy alignment efficiently. To design more human-like teachers, various human behaviors like stochasticity, myopic behavior, mistakes, etc. are integrated while generating preferences as in Lee et al. (2021); Park et al. (2022).

**Baselines:** We consider two state-of-the-art baselines for preference-based RL, which are PEBBLE (Lee et al., 2021) and PEBBLE+SURF (Park et al., 2022). We specifically compare with these two algorithms since PEBBLE+SURF (Park et al., 2022) and PEBBLE (Lee et al., 2021) already outperforms all the other existing algorithms, including (Metcalf et al., 2022; Christiano et al., 2017) in similar environment and configurations. It is important to note that PEBBLE+SURF utilizes additional unsupervised data and augmentations to improve the performance of PEBBLE, and hence depending on the quality and amount of the unsupervised information, the performance of PEBBLE+SURF varies rapidly. Hence, it is not extremely fair to compare PEBBLE+SURF with ours due to this obvious benefit, but still, we observe that our algorithm performs better than PEBBLE+SURF with controlled augmentations.

**Experimental Results Discussions and Evaluation:** To evaluate the performance of our algorithm against baselines, we select true episodic reward return as a valid metric. Since the eventual goal of any RL agent is to maximize the expected value of the episodic reward return, which is widely used in the literature. We conduct the experimental evaluations primarily centered on three key ideas - i.) First, we characterize the gap in the performance of the current SOTA methods due to inexact alignment strategies as demonstrated in Figure 1 which clearly shows a significant gap in current SOTA methods. ii.) Second, we compare the performance of our algorithm against baselines in the above benchmarks w.r.t ground truth return as shown in Figure 2 (averaged over 5 seeds). **Environments:** We evaluate the performance of the proposed algorithm for our framework PARL on large-scale continuous control robotics tasks in DM Control suite (Tassa et al., 2018) and Meta-World (Yu et al., 2021).

It demonstrates our algorithm's superiority over existing baselines in terms of episodic return, where A-PARL achieves near-oracle performance in a much faster time with an improved sample efficiency of approximately $63\%$. iii.) Finally, we test the alignment behavior of our learned policy by validating the generated trajectories on interacting with the environment against potential hacking or spurious learning. We validated if the agent is learning desired behaviors or has learned to hack the fitted reward due to the issue of reward overoptimization in RLHF (Gao et al., 2022) (specifically at an early stage of learning where it hits the high reward point). As observed in Figure 3, our agent learns the desired behavior and is able to open the door, whereas the existing algorithm (PEBBLE) fails to do so, demonstrating the effectiveness of our algorithm in alignment.

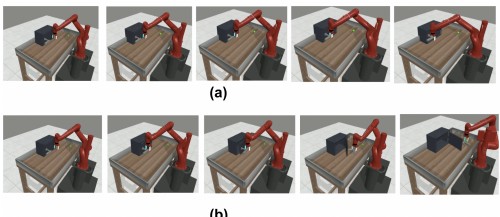

Figure 3: A visualization of learned behavior for the baseline Pebble (top row) and proposed (in the bottom row) (with policy at Env-Step $0.5 \times 10^6$). We note that the proposed A-PARL algorithm has been able to learn the aligned behavior of opening the door in the generated trajectory (top-right) whereas PEBBLE gets stuck depicting our algorithm's efficiency in alignment .

## 7 CONCLUSION

Potentially misaligned agents pose a severe risk to society; thus making AI alignment at the forefront of research of the current times (Liu et al., 2022). We identify a major gap within current algorithmic designs for AI alignment due to a lack of characterization of the dependence of the alignment objective on the data generated by policy trajectories. To mitigate the gap, we develop a unified bilevel optimization-based framework, PARL as a first step to address the policy alignment issue in reinforcement learning. Our proposed algorithm demonstrates superior performance over existing SOTA methods in large-scale robotics control tasks with provable convergence guarantees of rate $\mathcal{O}(1/T)$.

## 8 ACKNOWLEDGEMENT

Chakraborty and Huang are supported by the National Science Foundation NSF-IIS-FAI program, DOD-ONR-Office of Naval Research, DOD Air Force Office of Scientific Research, DOD-DARPA-Defense Advanced Research Projects Agency Guaranteeing AI Robustness against Deception (GARD), Adobe, Capital One, and JP Morgan faculty fellowships. Manocha and Bedi would like to acknowledge the support by Army Cooperative Agreement W911NF2120076 and Amazon Research Awards 2022.

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

# Appendix

## Table of Contents

# A  NOTATIONS

We collectively describe the notations used in this work in Table 1.

Table 1:

| Notations | Details |
|---|---|
| $\mathcal{S}, \mathcal{A}$ | State space, action space |
| $(s, a)$ | State-action pair |
| $P(s'\|s, a)$ | Transition kernel |
| $r_\nu(s, a)$ | Reward function parameterized by designer's parameters $\nu \in \mathbb{R}^n$ |
| $\pi_{\theta(\nu)}(\cdot\|s)$ | Policy parameterized by $\theta(\nu) \in \mathbb{R}^d$ for design parameters $\nu$ |
| $V_s(\nu, \theta(\nu))$ | lower objective - Value function for state $s$ at upper parameter $\nu$ and policy parameter $\theta(\nu)$ |
| $G(\nu, \theta^*(\nu))$ | upper objective |

# B  DETAILED CONTEXT OF RELATED WORK

**Bilevel Optimization.** Multi-stage optimization has a long-history in optimization and operations research, both for deterministic (Bertsimas & Caramanis, 2010) and stochastic objectives (Pflug & Pichler, 2014). In general, these problems exhibit NP-hardness if the objectives are general non-convex functions. Therefore, much work in this area restricts focus to classes of problems, such as affine (Bertsimas & Caramanis, 2010; Bertsimas et al., 2010). From both the mathematical optimization community (Bracken & McGill, 1973) and algorithmic game theory (Von Stackelberg, 2010), extensive interest has been given to specifically two-stage problems. A non-exhaustive list contains the following works (Ghadimi & Wang, 2018b; Yang et al., 2021; Khanduri et al., 2021; Li et al., 2022a; Ji & Liang, 2022; Huang et al., 2022). Predominately, one these works do not consider that either stage is a MDP, and therefore while some of the algorithmic strategies are potentially generalizable to the RL agent alignment problem, they are not directly comparable to the problem studied here.

**Stackleberg Games.** Algorithmic methods for Sackleberg games are a distinct line of inquiry that develops methods to reach the Stackleberg equilibrium of a game, which have received significant attention in recent years. Beginning with the static Stackleberg game setting, conditions for gradient play to achieve local equilibria have been established in (Fiez et al., 2019). Follow on work studied conditions for gradient play to converge without convexity, but does not allow for MDP/trajectory dependence of objective functions at either stage (Maheshwari et al., 2023). On the other hand, the access to information structures have gradually been relaxed to allow bandit feedback (Bai et al., 2021) and linear MDPs (Zhong et al., 2021). Value iteration schemes have been proposed to achieve Stackleberg-Nash equilibria as well (Goktas et al., 2022), although it is unclear how to generalize them to handle general policy parameterization in a scalable manner. Most similar to our work are those that develop implicit function-theorem based gradient play (Fiez et al., 2020; Vu et al.); however, there are no performance certificates for these approaches.

**Mechanism Design.** In this line of research, one studies the interrelationship between the incentives of an individual economic actor and their macro-level behavior at the level of a social welfare objective. This literature can be traced back to (Myerson, 1989; Hurwicz, 2003; Maskin, 2008), and typically poses the problem as one that does not involve sequential interactions. More recently, efforts to cast the evolution of the upper-stage which quantifies social welfare or ethical considerations as a sequential process, i.e., an MDP, have been considered (Tang, 2017; Hu et al., 2018). In these works, agents' behavior is treated as fixed and determining of the state transition dynamics, which gives rise to a distinct subclass of policy optimization problems (Lyu et al., 2022a;b).

**Reinforcement Learning with Preferences.** Revealed preferences is a concept in economics that states that humans in many cases do not know what they prefer until they are exposed to examples (Caplin & Dean, 2015). This idea gained traction as it provided a substantive basis for querying

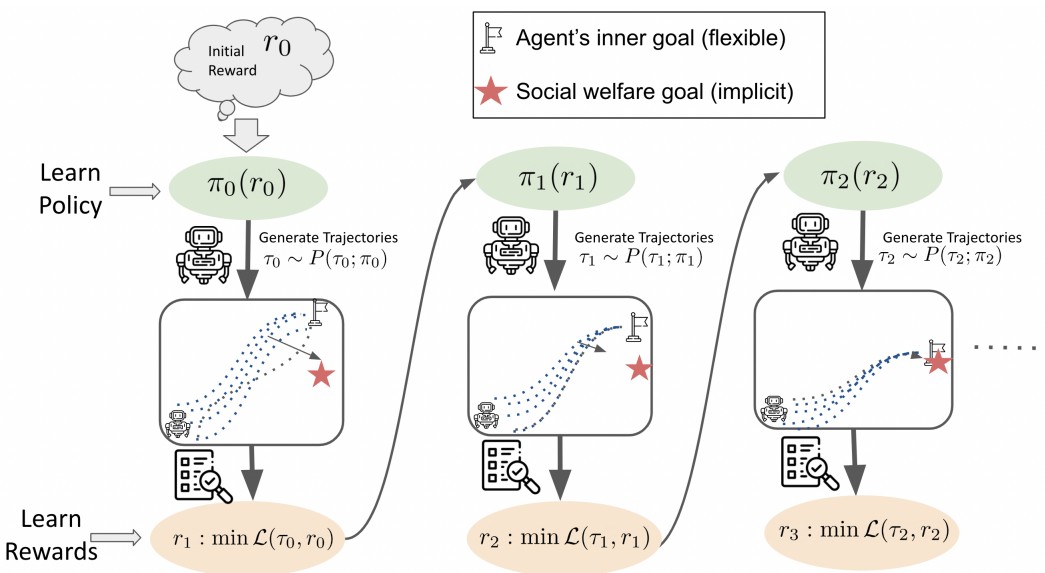

Figure 4: This figure describes the implementation flowchart of the iterative process of policy alignment in reinforcement learning. We start with some initial reward $r_0$, learn an optimal policy $\pi_0$ for that particular reward function at instant $t = 0$, and utility evaluates the policy to generate an updated reward function $r_1$. Then at the next iterate $t = 1$, we learn $\pi_1$ and so on.

humans and elucidating feedback on decision-making problems with metrics whose quantification is not obvious, such as trustworthiness or fairness. In its early stages, (Wilson et al., 2012) proposed a Bayesian framework to learn a posterior distribution over the policies directly from preferences. However, (Fürnkranz et al., 2012) focused on learning a cost or utility function as a supervised learning objective to maximize the likelihood of the preferences and subsequently maximize the policy under the estimated utility. An alternative and interesting line of research involves dueling bandits, which focuses on the comparison of two actions/trajectories and aims to minimize regret based on pairwise comparisons (Dudík et al., 2015; Bengs et al., 2021; Lekang & Lamperski, 2019; Pacchiano et al., 2023). (Christiano et al., 2017) was one of the first to scale deep preference-based learning for large-scale continuous control tasks by learning a reward function aligned with expert preferences. This was later improved by introducing additional demonstrations (Ibarz et al., 2018) and non-binary rankings (Cao et al., 2020). Most recent works (Lee et al., 2021; Park et al., 2022) improve the efficiency of preference-based learning for large-scale environments with off-policy learning and pre-training. However, a rigorous mathematical formulation analysis of the problem is still missing from the literature with special emphasis on the alignment objective and evaluations.

**Inverse Reinforcement Learning & Behavioral Cloning.** Implicitly contained in the RL with preferences framework is the assumption that humans should design a reward function to drive an agent toward the correct behavior through its policy optimization process. This question of reward design has also been studied through an alternative lens in which one provides demonstrations or trajectories, and seeks to fit a reward function to represent this information succinctly. Inverse RL (IRL) is concerned with inferring the underlying reward function from a set of demonstrations by human experts. Pioneering work in IRL, notably by Ng and Russell Ng & Russell (2000), centers on the max-margin inverse reinforcement learning framework for reward function estimation. Another prominent direction by Ziebart et al. (2008b; 2010) introduced the Max Entropy IRL framework which excelled in handling noisy trajectories with imperfect behavior via adopting a probabilistic perspective for reward learning. However, a major drawback in all these prior methods was the inability to incorporate unsuccessful demonstration which was efficiently solved in Shiarlis et al. (2016) with a constraint optimization formulation, by also maximizing the between the empirical feature expectation of unsuccessful examples and the feature expectation learned from the data. Interestingly Ho et al. (2016) posed it in the min-max formulation where the objective is to maximize the optimal policy and minimize the divergence to the occupancy indicated by the current policy and expert trajectories.

Later, with Deep RL methods, Finn et al. (2016); Wulfmeier et al. (2016) extended the maximum entropy deep IRL to continuous action and state spaces. Although IRL methods are effective, they still are extremely dependent on high-quality demonstrations which are expensive and might be hard to obtain for real-world sequential decision-making problems. Hence, there has been recent research on reward shaping and information-directed approaches Chakraborty et al. (2023b); Weerakoon et al. (2022); Bedi et al. (2022); Chakraborty et al. (2023a; 2022) which provides principled methods to learn under sparsity. However, such methods fail to scale in large state space robotics environments.

## C    ADDITIONAL MOTIVATING EXAMPLES

**Example 1: Energy efficient and sustainable design for robotic manipulation.** Consider a robotic manipulation task where the objective of the agent is to learn an optimal policy to transport components from a fixed position to a target position $\nu := (x, y)$. On the other hand, the designer's objective is to select the work-bench position $\nu$ to minimize the energy consumption of the robotic arm during the transportation task. Hence, it naturally boils down to the following bilevel problem as

$$\max_{\nu:=(x,y)} \mathbb{E}_{\rho(\tau;\theta^*(\nu))} \left[ \sum_{h=1}^{H_u} -a_h w_h \mid a_h \sim \pi_{\theta^*(\nu)}(\cdot|s_h) \right] \tag{23}$$
$$\text{s.t. } \theta^*(\nu) := \arg\max_{\theta} \mathbb{E} \left[ \sum_{h=0}^{H_\ell-1} \gamma^h r_\nu(s_h, a_h) \mid a_h \sim \pi_\theta(a_h|s_h), s_0 = s \right],$$

where $\mathbb{E}_{\rho(\tau;\theta^*(\nu))}$ denotes the expectation with respect to the trajectories collected by the lower-level optimal policy $\pi_{\theta^*(\nu)}$. In (23), action $a_h$ denotes the angular acceleration of the robotic arm, the state is represented by $s_h = (\alpha_h, w_h)$, $\alpha_t$ is the discretized angle, and $w_h$ angular velocity of the robotic arm. we define the transitions as $(\alpha_{t+1}, w_{t+1}) = (\alpha_h + w_h, w_h + a_h)$. The reward of the lower objective $r_\nu(s_h, a_h) = -\lambda_1 \|s_h - \nu\|^2 - \lambda_2 \|w_h\|^2$, i.e., reward increases as the arm moves closer to the workbench with a controlled angular velocity. The upper objective focuses on minimizing the energy emission during transport and is thus entangled with the trajectories generated under the optimal policy obtained via the lower-level. Therefore, to see that the problem in (23) is a special case of (2), we note that the upper objective (cf. (3)) takes the special form as follows $G(\nu, \theta^*(\nu)) = U(\pi_{\theta^*(\nu)}) = \mathbb{E}_{\rho(\tau;\theta^*(\nu))} \left[ \sum_{h=1}^{H_u} -a_h w_h \mid a_h \sim \pi_{\theta^*(\nu)}(\cdot|s_h) \right]$, where $Z(\nu) = 0$ for this example.

**Example 2: Social welfare aligned tax design for households.** Consider the problem of tax design for individual households while remaining attuned to social welfare, motivated by (Hill et al., 2021). From the household's perspective, each household seeks to maximize its own utility $u_h$ based on the number of working hours, consumption of goods, and net worth. Let us denote the accumulated asset as state $s_h$. At each time step $h$, the household agent selects an action $a_h = (n_h, c_{i,h}), a_h \sim \pi_\theta(a_h|s_h)$, where $n_h$ is the number of hours worked, and $c_{i,h}$ is the consumption of good $i$ at a pre-tax price of $p_i$, and $\theta$ denotes the policy parameter. We denote $f(s_h)$ as the reward for the accumulative asset $s_h$, updated at each time step by $s_{h+1} = s_h + (1-x)wn_h - \sum_{i=1}^{M} c_{i,h}$ and $\nu = (x, y_i)$ is the income tax rate and consumption tax rate for good $i$. Here we note that $x$ is a uniform tax across all households, whereas $y_i$ is a household-specific tax rate. Then the household agent's utility at time step $h$ is given by the equation $u_h = f(s_h) - \gamma n_h^2 + \prod_{i=1}^{M} \left( \frac{c_{i,h}}{p_i(1+y_i)} \right)^{\nu_i}$, where the product term corresponds to Cobb-Douglas function (Roth et al., 2016). In contrast, the objective of the regulatory body or government (upper-level) is to maximize the social welfare $v_t$ by adjusting the tax rates $\nu$ based on the optimal policy of the household agent (lower level). Hence, the upper objective representing the social welfare is defined as $v_h = g(s_h) + \sum_{i=1}^{M} \frac{c_{i,h}}{1+y_i} + \psi \ln \left( \frac{\prod_{i=1}^{M} c_{i,t} y_i}{\prod_{i=1}^{M} (1+y_i)} + wx_n h \right)$, where $g(\cdot)$ is the reward for the accumulative asset, $\psi$ is a positive constant, and $w$ is the wage rate. The household agent follows a policy that maximizes its discounted cumulative reward, while the social planner aims to maximize the discounted total social welfare by tuning the tax rates $x$ and $y_i$.

Thus, the bilevel formulation is given by

$$\max_{\nu} \mathbb{E}_{\rho(\tau;\theta^*(\nu))} \left[ \sum_{h=0}^{H_u-1} v_\nu(s_h, a_h) \right] \tag{24}$$

$$\text{s.t. } \theta^*(\nu) := \arg \max_{\theta} \mathbb{E} \left[ \sum_{h=0}^{H_\ell-1} \gamma^h u_\nu(s_h, a_h) \,|\, a_h \sim \pi_\theta(a_h|s_h), s_0 = s \right],$$

where $\gamma$ is the discount rate, and $\theta^*(\nu)$ represents the optimal lower policy of the household agent, which maximizes its expected cumulative return over a time horizon $H_\ell$.

**Example 3: Cost-effective robot navigation with Human feedback.** Consider an $N \times N$ maze-world environment where the robot needs to navigate from a start state $s$ to a goal state $g$. The maze is represented by a grid with discrete positions $(i, j)$, and the robot can take four actions $\{\uparrow, \downarrow, \leftarrow, \rightarrow\}$. The agent receives a reward $R_g(\tau)$ on reaching the goal and the objective of the agent is to learn the optimal policy $\pi_{\theta^*}$ for goal reaching. The designer on the other hand, bears an additional cost of moving the goal position from the terminal state $(N - 1, N - 1)$ and hence need to optimize for a general utility metric considering both the objectives and can be formulated as the bilevel optimization as

$$\max_{g} -\lambda_1 \|g - (N - 1, N - 1)\|_2 + \lambda_2 \mathbb{E}_{\pi_{\theta^*(\nu)}}[R_g(\tau)] \tag{25}$$

$$\text{s.t. } \theta^*(g) := \arg \max_{\theta} \mathbb{E} \left[ \sum_{h=0}^{H_\ell-1} \gamma^h r_g(s_h, a_h) \,|\, a_h \sim \pi_\theta(a_h|s_h), s_0 = s \right],$$

where the reward function is characterized by the goal state $g$. The upper objective deals with finding a suitable position of the goal state that can optimize the sustainability metric and the lower objective deals with optimal policy and reward achieved under the same.

## D GRADIENT EVALUATIONS FOR RLHF

In this section, we demonstrate how our A-PARL algorithm is applicable to the Reinforcement Learning from Human preferences paradigm. We begin by writing the gradient of the upper-level preference objective in equation (7) as

$$\nabla_\nu G(\nu, \theta^*(\nu)) = \nabla_\nu \mathbb{E}[y \log P_\nu(\tau_0 > \tau_1) + (1 - y) \log P_\nu(\tau_0 < \tau_1)] \tag{26}$$

$$= \nabla_\nu \mathbb{E}[f_\nu(\tau_0, \tau_1, y)]$$

$$= \nabla_\nu \sum_{\widetilde{\tau}} f_\nu(\tau_0, \tau_1, y) \cdot \rho(y, \tau_0, \tau_1; \theta^*(\nu))$$

where, let's denote $f_\nu(\tau_0, \tau_1, y) = y \log P_\nu(\tau_0 > \tau_1) + (1 - y) \log P_\nu(\tau_0 < \tau_1)$ for simplicity of notation and as stated before $\widetilde{\tau} = (\tau_0, \tau_1, y)$. Now expanding upon the gradient in equation (30), we have

$$\nabla_\nu G(\nu, \theta^*(\nu)) = \mathbb{E}[\nabla_\nu[f_\nu(\tau_0, \tau_1, y)]] + \sum_{\tau'} f_\nu(\tau_0, \tau_1, y) \cdot \nabla_\nu \rho(y, \tau_0, \tau_1, \theta^*(\nu)) \tag{27}$$

where, we use the chain-rule and replace summation with expectation to get the equation (27) .

$$\nabla_\nu G(\nu, \theta^*(\nu)) = \mathbb{E}[\nabla_\nu[f(\tau_0, \tau_1, y, \nu)]] + \sum_{\widetilde{\tau}} f(\tau_0, \tau_1, y, \nu) \cdot \nabla_\nu \rho(y, \tau_0, \tau_1; \theta^*(\nu)) \tag{28}$$

$$= \underbrace{\mathbb{E}[\nabla_\nu[f(\tau_0, \tau_1, y, \nu)]]}_{\text{Term 1}} + \underbrace{\mathbb{E}[f(\tau_0, \tau_1, y, \nu) \cdot \nabla_\nu \log \rho(y, \tau_0, \tau_1; \theta^*(\nu))]}_{\text{Term 2}}$$

where, we divide and multiply the second term by the $\rho(y, \tau_0, \tau_1; \theta^*(\nu))$ to get the log gradient term inside expectation. Now, to compute the expression, we first compute the gradient of $f(\tau_0, \tau_1, y, \nu)$. From the definition in equation (6), we know that

$$P_\nu(\tau_0 > \tau_1) = \frac{\exp R_\nu(\tau_0)}{\exp R_\nu(\tau_0) + \exp R_\nu(\tau_1)} \tag{29}$$

$$\log P_\nu(\tau_0 > \tau_1) = R_\nu(\tau_0) - \log(\exp R_\nu(\tau_0) + \exp R_\nu(\tau_1))$$

$$\log P_\nu(\tau_1 > \tau_0) = R_\nu(\tau_1) - \log(\exp R_\nu(\tau_0) + \exp R_\nu(\tau_1))$$

Now, with the above equation, the expression of Term 1 in equation (28) can be expanded as

$$\mathbb{E}[\nabla_\nu f(\tau_0, \tau_1, y, \nu)] = \mathbb{E}[y\nabla_\nu \log P_\nu(\tau_0 > \tau_1) + (1 - y)\nabla_\nu \log P_\nu(\tau_0 < \tau_1)] \tag{30}$$

$$= \mathbb{E}[y\nabla_\nu R_\nu(\tau_0) + (1 - y)R_\nu(\tau_1) - \frac{\nabla_\nu R_\nu(\tau_0)\exp R_\nu(\tau_0) + \nabla_\nu R_\nu(\tau_1)\exp R_\nu(\tau_1)}{\exp R_\nu(\tau_0) + \exp R_\nu(\tau_1)}]$$

$$= \mathbb{E}[y\nabla_\nu R_\nu(\tau_0) + (1 - y)R_\nu(\tau_1) - \nabla_\nu R_\nu(\tau_0)P_\nu(\tau_0 > \tau_1) - \nabla_\nu R_\nu(\tau_1)P_\nu(\tau_1 > \tau_0)]$$

Note that this term can be easily estimated under any differentiable parametrization of the reward function $R_\nu$ Next, we move to term 2, where the trajectories $\tau_0, \tau_1$ are drawn from the distribution $\rho(\tau; \theta^*(\nu))$ parametrized by the policy $\pi(\theta^*(\nu))$. The Term 2 from equation (28) can be written as :

$$\mathbb{E}[f_\nu(\tau_0, \tau_1, y) \cdot \nabla_\nu \log \rho(y, \tau_0, \tau_1; \theta^*(\nu))] \tag{31}$$

$$= \mathbb{E}[f_\nu(\tau_0, \tau_1, y) \cdot \nabla_\nu \log \Big(h(y|\tau_0, \tau_1)\rho(\tau_0; \theta^*(\nu))\rho(\tau_1; \theta^*(\nu))\Big)]$$

$$= \mathbb{E}[f_\nu(\tau_0, \tau_1, y) \cdot \Big(\nabla_\nu \log \rho(\tau_0; \theta^*(\nu)) + \nabla_\nu \log \rho(\tau_1; \theta^*(\nu))\Big)]$$

Now, the term $\nabla_\nu \log \rho(\tau_1, \theta^*(\nu))$ can be expressed as exactly done in equation (8)

$$\nabla_\nu \log \rho(\tau_1; \theta^*(\nu)) = \sum_t \nabla_\nu \log \pi_{\theta^*(\nu)}(a_t|s_t) \tag{32}$$

Now, exactly following the similar steps as done from equation (33) to equation (36) we can rewrite the gradient in the context of Bilevel RLHF as

$$\nabla_\nu f_h(\theta^*(\nu)) = \nabla_\nu \theta^*(\nu)^T \nabla_\theta f_h(\theta^*(\nu)). \tag{33}$$

From the first order optimality condition for the lower level objective, it holds that

$$\nabla_\theta V_s(\nu, \theta^*(\nu)) = 0, \tag{34}$$

which is the gradient of lower-level objective with respect to parameter $\theta$ evaluated at the optimal $\theta^*(\nu)$. Now, differentiating again with respect to $\nu$ on both sides of (34), we obtain

$$\nabla^2_{\nu,\theta} V_s(\nu, \theta^*(\nu)) + \nabla_\nu \theta^*(\nu)\nabla^2_\theta V_s(\nu, \theta^*(\nu)) = 0. \tag{35}$$

The above expression would imply that we can write the final expression for the gradient in (33) as

$$\nabla_\nu f_h(\theta^*(\nu)) = -\nabla^2_{v,\theta} V_s(\nu, \theta^*(\nu))\nabla^2_\theta V_s(\nu, \theta^*(\nu))^{-1}\nabla_\theta f_h(\theta^*(\nu)). \tag{36}$$

Now, replacing this in Term 2 we will get the final expression. It is important to note that it requires the same gradient computations as shown in Section 4 for our generalized policy alignment algorithm.

# E ADDITIONAL LEMMAS

**Lemma 3** (Value function related upper bounds). *Under Assumptions 1 - 4, it holds that*

*(i) The second order Jacobian term $\nabla^2_{\nu,\theta} V_s(\nu_t, \theta^K(\nu_t))$ is bounded as $\|\nabla^2_{\nu,\theta} V_s(\nu_t, \theta^K(\nu_t))\| \leq H^2_\ell L_r B$, where $H_\ell$ is the horizon length for the lower level [cf. (2)], $L_r$ is the reward Lipschitz parameter [cf. Assumption 2], and B is the score function bound [cf. Assumption 3].*

*(ii) The Hessian of the value function is Lipschitz with parameter as*

$$\|\nabla^2_\theta V_s(\nu, \theta^*(\nu)) - \nabla^2_\theta V_s(\nu, \theta^K(\nu))\| \leq L'\|\theta^*(\nu) - \theta^K(\nu)\|, \tag{37}$$

*where, $L' = L_{f_1}\chi_1 \frac{H_\ell}{2} L_2 + L_{f_1}$ and $L_{f_1} = L_2 H^2_\ell R$. Here, $H_\ell$ is the horizon length at the lower level, $\chi_1$ is a constant defined in (77), R is the maximum reward (cf. Assumption 2), and $L_2$ is the Lipschitz parameter of the Hessian of the policy defined in Assumption 3.*

*(iii) The second order mixed jacobian term $\nabla^2_{\nu,\theta} V_s(\nu, \theta^K(\nu))$ is Lipschitz continuous w.r.t $\theta$ i.e*

$$\|\nabla^2_{\nu,\theta} V_s(\nu, \theta^*(\nu_t)) - \nabla^2_{\nu,\theta} V_s(\nu_t, \theta^K(\nu))\| \leq L''\|\theta^*(\nu) - \theta^K(\nu)\| \tag{38}$$

*where, $L'' = L_{f_3}\chi_2 \frac{H_\ell}{2} L_2 + L_{f_3}$ and $L_{f_3} = L_r L_1 H^2_\ell$. Here, $\chi_2$ is a constant defined in equation (85), and other constants are as defined in statement (ii).*

The proof of Lemma 3 is in Appendix G.2. To prove this result, we start by considering the value function expression, and evaluating it's gradient, Hessian, and Jacobians. After expanding each of them, we separate the reward and policy-related terms and then utilize the aforementioned assumptions to upper bound them, respectively.

**Lemma 4.** *Let us define the update direction associated with the gradient in (13) as*

$$\phi_1(\tau) := U(\tau) \cdot \sum_{t=0}^{H_u} [-\nabla^2_{v,\theta} V_s(\nu, \theta^*(\nu)) \nabla^2_\theta V_s(\nu, \theta^*(\nu))^{-1} \nabla_\theta f_h(\theta^*(\nu))], \tag{39}$$

*and* $\phi_2(\tau) := U(\tau) \cdot \sum_{t=0}^{H_u} [-\nabla^2_{v,\theta} V_s(\nu, \theta^K(\nu)) \nabla^2_\theta V_s(\nu, \theta^K(\nu))^{-1} \nabla_\theta f_h(\theta^K(\nu))]. \tag{40}$

*Under Assumptions 1 - 4, it holds that*

$$\|\mathbb{E}_{\tau \sim \rho(\tau; \theta^K(\nu))}[\phi_1(\tau) - \phi_2(\tau)]\| \le H_u^2 \tilde{u} \gamma_1 \|\theta^*(\nu) - \theta^K(\nu)\|, \tag{41}$$

*where* $\gamma_1 := \kappa L_1 + \frac{L_{\nu,\theta} L'}{l_\pi^2} + \frac{L''}{l_\pi}$ *. Here, $\tilde{u}$ is the upper bound of utility $U(\tau)$ defined in (4), $l_\pi, L_1$ are policy-related Lipschitz parameters (cf. Assumption 3), $L'$ and $L''$ as defined in Lemma 3, and $\kappa$ mixed condition number defined in equation (109) and $L_{\nu,\theta}$ upper-bound of the norm of second order mixed jacobian term, defined in equation (107)*

The proof of Lemma 4 is in Appendix H.6.

# F    PROOF OF THEOREM 1

Without loss of generality, for the analysis, we consider the algorithm updates as if we are minimizing at the upper and lower level both.

*Proof.* We begin by the smoothness assumption in the upper-level objective (cf. Assumption 18), which implies that

$$G[\nu_{t+1}, \theta^*(\nu_{t+1})] \le G(\nu_t, \theta^*(\nu_t)) + \langle \nabla_\nu G(\nu_t, \theta^*(\nu_t)), \nu_{t+1} - \nu_t \rangle + \frac{L_g}{2} \|\nu_{t+1} - \nu_t\|^2. \tag{42}$$

From the update of upper parameter $\nu_{t+1}$ (cf. (14)), we holds that

$$G(\nu_{t+1}, \theta^*(\nu_{t+1})) \le G(\nu_t, \theta^*(\nu_t)) + \langle \nabla_\nu G(\nu_t, \theta^*(\nu_t)), -\alpha_u \widetilde{\nabla}_\nu G(\nu_t, \theta^K(\nu_t)) \rangle \tag{43}$$
$$+ \frac{L_g \alpha_u^2}{2} \|\widetilde{\nabla}_\nu G(\nu_t, \theta^K(\nu_t))\|^2.$$

We add subtract the original gradient $\nabla_\nu G(\nu_t, \theta^*(\nu_t))$ [cf. (13)] in (43) as follows

$$G(\nu_{t+1}, \theta^*(\nu_{t+1})) \le G(\nu_t, \theta^*(\nu_t)) + \langle \nabla_\nu G(\nu_t, \theta^*(\nu_t)), -\alpha_u \nabla_\nu G(\nu_t, \theta^*(\nu_t)) \rangle \tag{44}$$
$$+ \alpha_u \langle \nabla_\nu G(\nu_t, \theta^*(\nu_t)), \nabla_\nu G(\nu_t, \theta^*(\nu_t)) - \widetilde{\nabla}_\nu G(\nu_t, \theta^K(\nu_t)) \rangle$$
$$+ \frac{L_g \alpha_u^2}{2} \|\nabla_\nu G(\nu_t, \theta^*(\nu_t)) + \widetilde{\nabla}_\nu G(\nu_t, \theta^K(\nu_t)) - \nabla_\nu G(\nu_t, \theta^*(\nu_t))\|^2$$
$$= G(\nu_t, \theta^*(\nu_t)) - \alpha_u \|\nabla_\nu G(\nu_t, \theta^*(\nu_t))\|^2 \tag{45}$$
$$+ \alpha_u \langle \nabla_\nu G(\nu_t, \theta^*(\nu_t)), \nabla_\nu G(\nu_t, \theta^*(\nu_t)) - \widetilde{\nabla}_\nu G(\nu_t, \theta^K(\nu_t)) \rangle$$
$$+ \frac{L_g \alpha_u^2}{2} \|\nabla_\nu G(\nu_t, \theta^*(\nu_t)) + \widetilde{\nabla}_\nu G(\nu_t, \theta^K(\nu_t)) - \nabla_\nu G(\nu_t, \theta^*(\nu_t))\|^2.$$

Using Peter-Paul inequality for the third term on the right hand side of (45), we get

$$G(\nu_{t+1}, \theta^*(\nu_{t+1})) \le G(\nu_t, \theta^*(\nu_t)) - \alpha_u \|\nabla_\nu G(\nu_t, \theta^*(\nu_t))\|^2 + \frac{\alpha_u}{2c_1} \|\nabla_\nu G(\nu_t, \theta^*(\nu_t))\|^2 \tag{46}$$
$$+ \frac{\alpha_u c_1}{2} \|\nabla_\nu G(\nu_t, \theta^*(\nu_t)) - \widetilde{\nabla}_\nu G(\nu_t, \theta^K(\nu_t))\|^2$$
$$+ \frac{L_g \alpha_u^2}{2} \|\nabla_\nu G(\nu_t, \theta^*(\nu_t)) + \widetilde{\nabla}_\nu G(\nu_t, \theta^K(\nu_t)) - \nabla_\nu G(\nu_t, \theta^*(\nu_t))\|^2.$$

where $c_1 \geq 0$. Next, after grouping the terms, we get

$$
\begin{aligned}
G(\nu_{t+1}, \theta^*(\nu_{t+1})) \leq & G(\nu_t, \theta^*(\nu_t)) - \alpha_u \left(1 - \frac{1}{2c_1}\right) \|\nabla_\nu G(\nu_t, \theta^*(\nu_t))\|^2 \\
& + \frac{\alpha_u c_1}{2} \|\nabla_\nu G(\nu_t, \theta^*(\nu_t)) - \widetilde{\nabla}_\nu G(\nu_t, \theta^K(\nu_t))\|^2 \\
& + L_g \alpha_u^2 \|\nabla_\nu G(\nu_t, \theta^*(\nu_t))\|^2 + L_g \alpha_u^2 \|\widetilde{\nabla}_\nu G(\nu_t, \theta^K(\nu_t)) - \nabla_\nu G(\nu_t, \theta^*(\nu_t))\|^2 \\
= & G(\nu_t, \theta^*(\nu_t)) - \alpha_u \left(1 - \frac{1}{2c_1} - L_g \alpha_u\right) \|\nabla_\nu G(\nu_t, \theta^*(\nu_t))\|^2 \\
& + \alpha_u \left(\frac{c_1}{2} + L_g \alpha_u\right) \|\nabla_\nu G(\nu_t, \theta^*(\nu_t)) - \widetilde{\nabla}_\nu G(\nu_t, \theta^K(\nu_t))\|^2, \quad (47)
\end{aligned}
$$

where we use the inequality $\|a + b\|^2 \leq 2\|a\|^2 + 2\|b\|^2$, followed by algebraic operations to get the final expression of equation (47). Next, we analyze the term $\|\nabla_\nu G(\nu_t, \theta^*(\nu_t)) - \widetilde{\nabla}_\nu G(\nu_t, \theta^K(\nu_t))\|^2$ from equation (47). It is important to note that the dependency of the lower optimal variable on the sampling distribution of the upper objective is novel compared to the standard class of stochastic bilevel problems and requires a novel analysis which we emphasize in the subsequent sections. Let us start by considering the explicit expressions of $\nabla_\nu G(\nu, \theta^*(\nu))$ and $\widetilde{\nabla}_\nu G(\nu_t, \theta^K(\nu_t))$ in equations (13) and (14) as

$$
\nabla_\nu G(\nu, \theta^*(\nu)) - \widetilde{\nabla}_\nu G(\nu, \theta^K(\nu)) = \mathbb{E}_{\tau \sim \rho(\tau; \theta^*(\nu))}[\phi_1(\tau)] - \mathbb{E}_{\tau \sim \rho(\tau; \theta^K(\nu))}[\phi_2(\tau)], \quad (48)
$$

where we define

$$
\phi_1(\tau) = U_\nu(\tau) \cdot \sum_{t=0}^{H_u} [-\nabla_{v,\theta}^2 V_s(\nu, \theta^*(\nu)) \nabla_\theta^2 V_s(\nu, \theta^*(\nu))^{-1} \nabla_\theta f_h(\theta^*(\nu))] + \nabla_\nu U_\nu(\tau), \quad (49)
$$

$$
\text{and } \phi_2(\tau) = U_\nu(\tau) \cdot \sum_{t=0}^{H_u} [-\nabla_{v,\theta}^2 V_s(\nu, \theta^K(\nu)) \nabla_\theta^2 V_s(\nu, \theta^K(\nu))^{-1} \nabla_\theta f_h(\theta^K(\nu))] + \nabla_\nu U_\nu(\tau). \quad (50)
$$

Now, we expand the terms in (48) by adding subtracting the term $\mathbb{E}_{\tau \sim \rho(\tau; \theta^K(\nu))}[\phi_1(\tau)]$ in the right hand side as follows

$$
\begin{aligned}
\nabla_\nu G(\nu, \theta^*(\nu)) - \widetilde{\nabla}_\nu G(\nu, \theta^K(\nu)) = & \mathbb{E}_{\tau \sim \rho(\tau; \theta^*(\nu))}[\phi_1(\tau)] - \mathbb{E}_{\tau \sim \rho(\tau; \theta^K(\nu))}[\phi_1(\tau)] \quad (51) \\
& + \mathbb{E}_{\tau \sim \rho(\tau; \theta^K(\nu))}[\phi_1(\tau)] - \mathbb{E}_{\tau \sim \rho(\tau; \theta^K(\nu))}[\phi_2(\tau)] \\
= & \mathbb{E}_{\tau \sim \rho(\tau; \theta^*(\nu))}[\phi_1(\tau)] - \mathbb{E}_{\tau \sim \rho(\tau; \theta^K(\nu))}[\phi_1(\tau)] \\
& + \mathbb{E}_{\tau \sim \rho(\tau; \theta^K(\nu))}[\phi_1(\tau) - \phi_2(\tau)]. \quad (52)
\end{aligned}
$$

This is an extremely critical point of departure from existing bilevel algorithms where we characterize the complexity in the sampling distribution (mentioned above) by breaking into 2 parts and introducing a notion of probabilistic divergence in the analysis. We note that the first term on the right-hand side of (52) can be written as :

$$
\mathbb{E}_{\tau \sim \rho(\tau; \theta^*(\nu))}[\phi_1(\tau)] - \mathbb{E}_{\tau \sim \rho(\tau; \theta^K(\nu))}[\phi_1(\tau)] \leq \sup_{\phi \in \mathcal{F}} \mathbb{E}_{\tau \sim \rho(\tau; \theta^*(\nu))}[\phi_1(\tau)] - \mathbb{E}_{\tau \sim \rho(\tau; \theta^K(\nu))}[\phi_1(\tau)]
$$
$$
(53)
$$

This boils down to the standard definition of Integral Probability Metric(IPM) which, under suitable assumption on the function class $\mathcal{F}$ can generalize various popular distance measures in probability theory for ex : Total variation, Wasserstein, Dudley metric etc. For a more detailed discussion on the connection to f-divergence and IPM, refer to (Sriperumbudur et al., 2009). Specifically, for our analysis, we will be dealing primarily with a subclass of f-divergences that satisfy the triangle inequality as $D_f(p, q) \leq D_f(p, r) + D_f(r, q)$ which includes divergences such as Total variation or Hellinger distances. Specifically, we show that under certain boundedness conditions on the function class $\phi \in \mathcal{F} : \|\phi\| \leq \zeta$, we can upper-bound the divergence which is a critical and interesting point of our analysis. Interestingly for the Reinforcement learning problem under study, $\phi$ satisfies such a boundedness condition which we prove in sections. Thus we were able to utilize the special structure in RL to solve this special type of Bilevel problems.

On the other-hand, the second term on the right-hand side of (52) is an expected difference between the two functions. Hence, we can write (52) as

$$\nabla_\nu G(\nu, \theta^*(\nu)) - \widetilde{\nabla}_\nu G(\nu, \theta^K(\nu)) = D_f(\rho(\tau; \theta^*(\nu)), \rho(\tau; \theta^K(\nu)))$$
$$+ \mathbb{E}_{\tau \sim \rho(\tau; \theta^K(\nu))}[\phi_1(\tau) - \phi_2(\tau)], \tag{54}$$

where $D_f$ denotes the f-divergence between two distributions. Taking the norm on both sides and from the statements of Lemma 1 and Lemma 4, we can write

$$\|\nabla_\nu G(\nu, \theta^*(\nu)) - \widetilde{\nabla}_\nu G(\nu, \theta^K(\nu))\|^2 \leq \left(\frac{H_u^2 L_2^2}{2} + 2H_u^4 \tilde{u}^2 \gamma_1^2\right) \|\theta^*(\nu) - \theta^K(\nu)\|^2. \tag{55}$$

Utilizing this bound in (47), we get

$$G(\nu_{t+1}, \theta^*(\nu_{t+1})) - G(\nu_t, \theta^*(\nu_t))$$
$$\leq -\alpha_u \left(1 - \frac{1}{2c_1} - L_g \alpha_u\right) \|\nabla_\nu G(\nu_t, \theta^*(\nu_t))\|^2$$
$$+ \alpha_u \left(\frac{c_1}{2} + L_g \alpha_u\right) \left(\frac{H_u^2 L_2^2}{2} + 2H_u^4 \tilde{u}^2 \gamma_1^2\right) \|\theta^*(\nu) - \theta^K(\nu)\|^2, \tag{56}$$

where we write the final expression for the convergence analysis from equation (47) and for simplicity of notations, let's assume $\delta_1 = \alpha_u \left(1 - \frac{1}{2c_1} - L_g \alpha_u\right)$ and $\delta_2 = \alpha_u \left(\frac{c_1}{2} + L_g \alpha_u\right) \left(\frac{H_u^2 L_2^2}{2} + 2H_u^4 \tilde{u}^2 \gamma_1^2\right)$, which leads to the simplified version of the equation (56)

$$G(\nu_{t+1}, \theta^*(\nu_{t+1})) - G(\nu_t, \theta^*(\nu_t)) \leq -\delta_1 \|\nabla_\nu G(\nu_t, \theta^*(\nu_t))\|^2 + \delta_2 \|\theta^*(\nu_t) - \theta^K(\nu_t)\|^2. \tag{57}$$

From the statement of Lemma 2, we can upper bound the above expression as

$$G(\nu_{t+1}, \theta^*(\nu_{t+1})) - G(\nu_t, \theta^*(\nu_t)) \leq -\delta_1 \|\nabla_\nu G(\nu_t, \theta^*(\nu_t))\|^2 + \delta_2 \frac{\eta^K L_6}{\mu} Z, \tag{58}$$

where we know $\eta \in (0, 1)$. Next, we select $K = t + 1$ to obtain

$$G(\nu_{t+1}, \theta^*(\nu_{t+1})) - G(\nu_t, \theta^*(\nu_t)) \leq -\delta_1 \|\nabla_\nu G(\nu_t, \theta^*(\nu_t))\|^2 + \delta_2 \frac{\eta^{t+1} L_6}{\mu} Z. \tag{59}$$

Taking the summation over $t = 0$ to $T - 1$ on both sides, we get

$$G(\nu_T, \theta^*(\nu_T) - G(\nu_0, \theta^*(\nu_0)) \leq -\delta_1 \sum_{t=0}^{T-1} \|\nabla_\nu G(\nu_t, \theta^*(\nu_t))\|^2 + \frac{\delta_2 L_6 Z}{\mu} \sum_{t=0}^{T-1} \eta^{t+1} \tag{60}$$

After rearranging the terms, we get

$$\sum_{t=0}^{T-1} \|\nabla_\nu G(\nu_t, \theta^*(\nu_t))\|^2 \leq \frac{G(\nu_0, \theta^*(\nu_0)) - G(\nu_T, \theta^*(\nu_T)}{\delta_1} + \frac{\eta \delta_2 L_6 Z}{\delta_1 \mu} \sum_{t=0}^{T-1} \eta^t$$
$$\leq \frac{G(\nu_0, \theta^*(\nu_0)) - G(\nu_T, \theta^*(\nu_T)}{\delta_1} + \frac{\eta \delta_2 L_6 Z}{\delta_1 \mu (1 - \eta)}. \tag{61}$$

Let us denote $G_0 := G(\nu_0, \theta^*(\nu_0))$ and upper bound $-G(\nu_T, \theta^*(\nu_T)) \leq -G^*$ where $G^*$ denotes the global optimal value of the upper objective. After dividing both sides in (61) by $T$, we get

$$\frac{1}{T} \sum_{t=0}^{T-1} \|\nabla_\nu G(\nu_t, \theta^*(\nu_t))\|^2 \leq \frac{G_0 - G^*}{\delta_1 T} + \frac{\eta \delta_2 L_6 Z}{T \delta_1 \mu (1 - \eta)}. \tag{62}$$

□

# G    Proof of Lemmas

## G.1    Proof of Lemma 1

*Proof.* The probability distribution of the trajectory $\tau = \{s_h, a_h\}_{h=1}^H$ is given by (as defined after the equation (3) in main body)

$$\rho(\tau; \theta^*(\nu)) = \rho(s_0) \prod_{h=1}^{H} \pi_{\theta^*(\nu)}(a_h|s_h) P(s_{h+1}|s_h, a_h). \tag{63}$$

Similarly, we can derive an equivalent expression for the probability of trajectory induced by the policy $\pi_{\theta^K(\nu)}$ by replacing $\theta^*(\nu)$ with $\theta^K(\nu)$. Here, $P(s_{h+1}|s_h, a_h)$ is the transition probability which remains the same for both and $\rho(s_0)$ is the initial state distribution. Next, the f-divergence between the two distributions $D_f(\rho(\tau; \theta^*(\nu)), \rho(\tau; \theta^K(\nu)))$ can be written as

$$D_f(\rho(\tau; \theta^*(\nu)), \rho(\tau; \theta^K(\nu))) \leq \underbrace{D_f(\rho(\tau; \theta^*(\nu)), \rho(\tau; \beta))}_{I} + \underbrace{D_f(\rho(\tau; \beta), \rho(\tau; \theta^K(\nu)))}_{II}, \tag{64}$$

which holds by triangle inequality of the class of f-divergences we considered as discussed above. $\rho(\tau; \beta)$ represents the trajectory probability induced by another hybrid policy $\pi_\beta(\cdot|s)$ which executes the action based on the policy $\pi_{\theta^K(\nu)}(\cdot|s)$ for the first time-step and then follows the policy $\pi_{\theta^*(\nu)}(\cdot|s)$ for subsequent timesteps. Now, we focus on term I in (64), we get

$$D_f(\rho(\tau; \theta^*(\nu)), \rho(\tau; \beta))$$

$$= \sum_\tau \rho(\tau; \beta) f\left(\frac{\rho(\tau; \theta^*(\nu))}{\rho(\tau; \beta)}\right) \tag{65}$$

$$= \sum_\tau \rho(\tau; \beta)) f\left(\frac{\rho(s_0)\pi_{\theta^*(\nu)}(a_0|s_0)P(s_1|s_0, a_0)\prod_{h=1}^{H}\pi_{\theta^*(\nu)}(a_h|s_h)P(s_{h+1}|s_h, a_h)}{\rho(s_0)\pi_{\theta^K(\nu)}(a_0|s_0)P(s_1|s_0, a_0)\prod_{h=1}^{H}\pi_{\theta^*(\nu)}(a_h|s_h)P(s_{h+1}|s_h, a_h)}\right)$$

$$= \sum_\tau \rho(\tau; \beta)) f\left(\frac{\pi_{\theta^*(\nu)}(a_0|s_0)}{\pi_{\theta^K(\nu)}(a_0|s_0)}\right),$$

where first we expand upon the definition of the trajectory distribution induced by both policies and get the final expression of the equation (65). By expanding the term $\rho(\tau; \beta)$ in (65), we obtain

$$D_f(\rho(\tau; \theta^*(\nu)), \rho(\tau; \beta)) = \sum_{s_0} \rho(s_0) \sum_{a_0} \pi_{\theta^K(\nu)}(a_0|s_0) f\left(\frac{\pi_{\theta^*(\nu)}(a_0|s_0)}{\pi_{\theta^K(\nu)}(a_0|s_0)}\right) \sum_{s_1} P(s_1|s_0, a_0) \cdots$$

$$= \sum_s \rho(s) \sum_a \pi_{\theta^K(\nu)}(a|s) f\left(\frac{\pi_{\theta^*(\nu)}(a|s)}{\pi_{\theta^K(\nu)}(a|s)}\right)$$

$$= \mathbb{E}_{\rho(s)} \sum_a \pi_{\theta^K(\nu)}(a|s) f\left(\frac{\pi_{\theta^*(\nu)}(a|s)}{\pi_{\theta^K(\nu)}(a|s)}\right)$$

$$= \mathbb{E}_{\rho(s)}[D_f(\pi_{\theta^*(\nu)}(a|s), \pi_{\theta^K(\nu)}(a|s))], \tag{66}$$

where, in the first equation we expand upon the sum over all trajectories with the occupancy distribution over states and actions, and replacing with f-divergence, we get the final expression.

Next, we expand similarly for the term II in (64) and expand as

$$D_f(\rho(\tau;\beta), \rho(\tau;\theta^K(\nu)))$$

$$= \sum_\tau \rho(\tau;\theta^K(\nu)))f\left(\frac{\rho(\tau;\beta)}{\rho(\tau;\theta^K(\nu))}\right)$$

$$= \sum_\tau \rho(\tau;\theta^K(\nu)))f\left(\frac{\rho(s_0)\pi_{\theta^K(\nu)}(a_0|s_0)P(s_1|s_0,a_0)\prod_{h=1}^{H}\pi_{\theta^*(\nu)}(a_h|s_h)P(s_{h+1}|s_h,a_h)}{\rho_0(s_0)\pi_{\theta^K(\nu)}(a_0|s_0)P(s_1|s_0,a_0)\prod_{h=1}^{H}\pi_{\theta^K(\nu)}(a_h|s_h)P(s_{h+1}|s_h,a_h)}\right)$$

$$= \sum_{s_0}\rho(s_0)\sum_{a_1}\pi_{\theta^K(\nu)}(a_0|s_0)\sum_{s_1}P(s_1|s_0,a_0)\cdots f\left(\frac{\prod_{h=1}^{H}\pi_{\theta^*(\nu)}(a_h|s_h)P(s_{h+1}|s_h,a_h)}{\prod_{h=1}^{H}\pi_{\theta^K(\nu)}(a_h|s_h)P(s_{h+1}|s_h,a_h)}\right)$$

$$= \sum_{s_0}\rho(s_0)\sum_{a_1}\pi_{\theta^K(\nu)}(a_1|s_0)\sum_{\tau_1}\rho(\tau_1;\theta^K(\nu), \frac{\rho(\tau_1;\theta^*(\nu)}{\rho(\tau_1;\theta^K(\nu))}$$

$$= \sum_{s_0}\rho(s_0)\sum_{a_1}\pi_{\theta^K(\nu)}(a_1|s_0)D_f(\rho(\tau_1;\theta^*(\nu), \rho(\tau_1;\theta^K(\nu)), \tag{67}$$

where we expand the trajectory distribution induced by the policies and subsequently express as the ratio of the probability of trajectories wrt $\tau_1$, we get the final expression. Now, we expand upon the f-divergence of the trajectory $\tau_1$ distribution as

$$D_f(\rho(\tau;\beta), \rho(\tau;\theta^K(\nu))) = \sum_{s_0}\rho(s_0)\sum_{a_1}\pi_{\theta^K(\nu)}(a_1|s_0)D_f(\rho(\tau_1;\theta^*(\nu), \rho(\tau_1;\theta^K(\nu)) \tag{68}$$

$$\leq \sum_{s_0}\rho(s_0)\sum_{a_1}\pi_{\theta^K(\nu)}(a_1|s_0)\Big(D_f(\rho(\tau_1;\theta^*(\nu)), \rho(\tau_1;\beta))$$

$$+ D_f(\rho(\tau_1;\beta), \rho(\tau_1;\theta^K(\nu)))\Big),$$

where using the triangle inequality and get back the similar form with which we had started in equation (64) similar to term I and term II. Here, similarly continuing this expansion, we finally get

$$D_f(\rho(\tau;\theta^*(\nu)), \rho(\tau;\theta^K(\nu))) \leq \sum_{h=0}^{H-1}\mathbb{E}_{s\sim\rho_{\theta^K(\nu)}^P(s)}D_f(\pi_{\theta^*(\nu)}(a|s), \pi_{\theta^K(\nu)}(a|s))$$

$$\leq H\mathbb{E}_{s\sim\rho_{\theta^K(\nu)}(s)}D_f(\pi_{\theta^*(\nu)}(a|s), \pi_{\theta^K(\nu)}(a|s))$$

$$\leq H\sum_s\rho_{\theta^K(\nu)}(s)D_f(\pi_{\theta^*(\nu)}(a|s), \pi_{\theta^K(\nu)}(a|s))$$

$$\leq HD_f(\pi_{\theta^*(\nu)}(a'|s'), \pi_{\theta^K(\nu)}(a'|s')), \tag{69}$$

where, we upper bound the first equation by the total number of timesteps or horizon length $H$ of the trajectory and subsequently upper-bound the divergence by the state $(s,a)$ pair for which the $D_f(\pi_{\theta^*(\nu)}(a|s), \pi_{\theta^K(\nu)}(a|s))$ is maximum and is given as $(s', a')$. Next, in (69), by considering the total variation as the f-divergence (it falls into the class of f-divergence under consideration) and expanding using definition with countable measures to obtain

$$D_f(\rho(\tau;\theta^*(\nu)), \rho(\tau;\theta^K(\nu))) \leq HD_{TV}(\pi_{\theta^*(\nu)}(a'|s'), \pi_{\theta^K(\nu)}(a'|s'))$$

$$\leq \frac{H}{2}\|\pi_{\theta^*(\nu)}(a'|s') - \pi_{\theta^K(\nu)}(a'|s')\|_1$$

$$\leq \frac{HL_2}{2}\|\theta^*(\nu) - \theta^K(\nu)\|, \tag{70}$$

where, we use the gradient-boundedness condition on the score function (cf. Assumption 3) on the policy parameter to get the final expression of equation (70). We note that the result holds for any general horizon length $H$. $\qquad\square$

### G.2 Proof of Lemma 3

#### G.2.1 Proof of Lemma 3 Statement (i)

*Proof.* We start with the definition from (17)

$$\nabla^2_{\nu,\theta} V_s(\nu_t, \theta^K(\nu_t)) = \sum_\tau \rho(\tau; \theta^K(\nu_t)) \left[ \sum_{h=0}^{H_\ell} \gamma^h \cdot \nabla_\nu r_{\nu_t}(s_h, a_h) \cdot \left( \sum_{j=0}^h \nabla_\theta \log \pi_{\theta^K(\nu_t)}(a_j|s_j) \right)^T \right]$$

$$\leq \left[ \sum_{h=0}^{H_\ell} \gamma^h \cdot \nabla_\nu r_{\nu_t}(s'_h, a'_h) \cdot \left( \sum_{j=0}^h \nabla_\theta \log \pi_{\theta^K(\nu_t)}(a'_j|s'_j) \right)^T \right] \sum_\tau \rho(\tau; \theta^K(\nu_t))$$

$$= \left[ \sum_{h=0}^{H_\ell} \gamma^h \cdot \nabla_\nu r_{\nu_t}(s'_h, a'_h) \cdot \left( \sum_{j=0}^h \nabla_\theta \log \pi_{\theta^K(\nu_t)}(a'_j|s'_j) \right)^T \right], \tag{71}$$

where $\tau' = [(s'_0, a'_0) \cdot (s'_h, a'_h) \cdots]$ represents the trajectory for which the lower product term is maximum and thereby upper-bounding with that leads to expression in equation (71). Next we upper-bound the norm of the $\|\nabla^2_{\nu,\theta} V_s(\nu_t, \theta^K(\nu_t))\|$ as

$$\|\nabla^2_{\nu,\theta} V_s(\nu_t, \theta^K(\nu_t))\| \leq \left\| \sum_{h=0}^{H_\ell} \gamma^h \cdot \nabla_\nu r_{\nu_t}(s'_h, a'_h) \cdot \left( \sum_{j=0}^h \nabla_\theta \log \pi_{\theta^K(\nu_t)}(a'_j|s'_j) \right)^T \right\|$$

$$\leq \left\| \sum_{h=0}^{H_\ell} \gamma^h \cdot \nabla_\nu r_{\nu_t}(s'_h, a'_h) \right\| \cdot \left\| \sum_{j=0}^h \nabla_\theta \log \pi_{\theta^K(\nu_t)}(a'_j|s'_j) \right\|$$

$$\leq H_\ell^2 L_r B, \tag{72}$$

where we apply the Cauchy-Schwarz inequality to get the inequality in the second line. Subsequently, we apply triangle inequality with reward boundedness assumption from Assumption 2 and bounded score function of the policy from Assumption 3 to get the final bound for the mixed hessian term. □

#### G.2.2 Proof of Lemma 3 Statement (ii)

*Proof.* We start by considering the term

$$\nabla^2_\theta V_s(\nu, \theta^*(\nu)) - \nabla^2_\theta V_s(\nu, \theta^K(\nu)) = \mathbb{E}_{\rho(\tau; \theta^*(\nu))} f_1(\tau) - \mathbb{E}_{\rho(\tau; \theta^K(\nu))} f_2(\tau), \tag{73}$$

where we define

$$f_1(\tau) = \sum_{h=0}^{H_\ell} \gamma^{h-1} \cdot r_{\nu_t}(s_h, a_h) \cdot \left( \sum_{j=0}^h \nabla^2_\theta \log \pi_{\theta^*(\nu)}(a_j|s_j) \right) \tag{74}$$

$$f_2(\tau) = \sum_{h=0}^{H_\ell} \gamma^{h-1} \cdot r_{\nu_t}(s_k, a_k) \cdot \left( \sum_{j=0}^h \nabla^2_\theta \log \pi_{\theta^K(\nu)}(a_j|s_j) \right). \tag{75}$$

Subsequently, we write the norm of the equation (73) into 2 parts as

$$\|\nabla^2_\theta V_s(\nu, \theta^*(\nu)) - \nabla^2_\theta V_s(\nu, \theta^K(\nu))\|$$
$$= \|\mathbb{E}_{\rho(\tau; \theta^*(\nu))} f_1(\tau) - \mathbb{E}_{\rho(\tau; \theta^K(\nu))} f_1(\tau) + \mathbb{E}_{\rho(\tau; \theta^K(\nu))} f_1(\tau) - \mathbb{E}_{\rho(\tau; \theta^K(\nu))} f_2(\tau)\|$$
$$\leq \|T_1\| + \|T_2\|, \tag{76}$$

First we use triangle inequality and then add and subtract $\mathbb{E}_{\rho(\tau; \theta^K(\nu))} f_1(\tau)$ to get the expression where, $T_1 = \mathbb{E}_{\rho(\tau; \theta^*(\nu))} f_1(\tau) - \mathbb{E}_{\rho(\tau; \theta^K(\nu))} f_1(\tau)$ and $T_2 = \mathbb{E}_{\rho(\tau; \theta^K(\nu))} f_1(\tau) - \mathbb{E}_{\rho(\tau; \theta^K(\nu))} f_2(\tau)$. This is an interesting point of departure from existing Bilevel analysis where the stochasticity is mainly iid noise. Hence to deal with this, we separate the terms 1. divergence b/w two probability distributions and 2. expected diff b/w two functions under the same distribution. We next, upper-bound the individual terms to get the final Lispchitz constant.

First, we focus on the first term of inequality i.e $T_1$ given as

$$T_1 = \mathbb{E}_{\rho(\tau;\theta^*(\nu))} f_1(\tau) - \mathbb{E}_{\rho(\tau;\theta^K(\nu))} f_1(\tau) \tag{77}$$
$$\leq \sup_{f_1} [\mathbb{E}_{\rho(\tau;\theta^*(\nu))} f_1(\tau) - \mathbb{E}_{\rho(\tau;\theta^K(\nu))} f_1(\tau)]$$
$$\leq \chi_1 d_{TV}(\rho(\tau;\theta^*(\nu)), \rho(\tau;\theta^K(\nu)))$$
$$\leq \chi_1 \frac{H_\ell}{2} L_2 \|\theta^*(\nu) - \theta^K(\nu)\|,$$

where we first upper-bound the expression to a standard Integral Probability Metric form by taking the supremum and then dividing and multiplying with the norm of the function given by $\chi_1 = \|f_1(\cdot)\| = H_l^2 R L_1$ from equation (96). With that, we get the expression as the of Total variation distance(by definition (Sriperumbudur et al., 2009)) in the third inequality. Then, we upper-bounded the total variation using the results from equation (70) to get the final expression. Note that, in equation (70), we have shown for any general $H$, which will be $H_l$ for our case.

Now, we proceed to the second term of the equation $T_2$ and derive an upper bound as

$$T_2 = \sum_\tau \rho(\tau;\theta^K(\nu))(f_1(\tau) - f_2(\tau)) \tag{78}$$
$$\leq f_1(\tau') - f_2(\tau')$$
$$= \sum_{h=0}^{H_\ell} \gamma^{h-1} \cdot r_{\nu_t}(s_h, a_h) \cdot \left( \sum_{j=0}^{h} (\nabla_\theta^2 \log \pi_{\theta^*(\nu)}(a_j|s_j) - \nabla_\theta^2 \log \pi_{\theta^K(\nu)}(a_j|s_j)) \right)$$

,

where we consider the trajectory $\tau'$ in the sum with the maximum value and upper bound by that and expand the definition of $f_1, f_2$ to get expression in equation (78).

$$\|T_2\| \leq \| \sum_{h=0}^{H_\ell} \gamma^{h-1} \cdot r_{\nu_t}(s_h, a_h) \cdot \left( \sum_{j=0}^{h} (\nabla_\theta^2 \log \pi_{\theta^*(\nu)}(a_j|s_j) - \nabla_\theta^2 \log \pi_{\theta^K(\nu)}(a_j|s_j)) \right) \| \tag{79}$$
$$\leq \| \sum_{h=0}^{H_\ell} \gamma^{h-1} \cdot r_{\nu_t}(s_h, a_h) \cdot \left( \sum_{j=0}^{h} (\nabla_\theta^2 \log \pi_{\theta^*(\nu)}(a_j|s_j) - \nabla_\theta^2 \log \pi_{\theta^K(\nu)}(a_j|s_j)) \right) \|$$
$$\leq \sum_{h=0}^{H_\ell} \gamma^{h-1} \cdot \|r_{\nu_t}(s_h, a_h)\| \left( \sum_{j=0}^{h} (\|\nabla_\theta^2 \log \pi_{\theta^*(\nu)}(a_j|s_j) - \nabla_\theta^2 \log \pi_{\theta^K(\nu)}(a_j|s_j)\|) \right)$$
$$\leq \sum_{h=0}^{H_\ell} \gamma^{h-1} \cdot \|r_{\nu_t}(s_h, a_h)\| H_\ell L_2 \|\theta^*(\nu) - \theta^K(\nu)\|$$
$$= L_2 R H_\ell^2 \|\theta^*(\nu) - \theta^K(\nu)\|,$$

where we use Cauchy-Schwartz and triangle inequality repetitively to get to the third inequality. Next, we use Assumption 3 on the Hessian Lipschitzness of the score function and the bounded reward norm $\max_{(s,a)} \|r_\nu(s,a)\| = R$ to get the next inequality. Finally, we use the upper bound on the geometric series to obtain the final expression. Adding equations (77) and (79), we get the

$$\|\nabla_\theta^2 V_s(\nu, \theta^*(\nu)) - \nabla_\theta^2 V_s(\nu, \theta^K(\nu))\| \leq L' \|\theta^*(\nu) - \theta^K(\nu)\| \tag{80}$$

where, $L' = L_1 L_2 R \frac{H_l^3}{2} + L_2 R H_\ell^2$ . □

### G.2.3 Proof of Lemma 3 Statement (iii)

*Proof.* We start by considering the term

$$\nabla_{\nu,\theta}^2 V_s(\nu, \theta^*(\nu)) - \nabla_{\nu,\theta}^2 V_s(\nu, \theta^K(\nu)) \leq \mathbb{E}_{\rho(\tau;\theta^*(\nu))} f_3(\tau) - \mathbb{E}_{\rho(\tau;\theta^K(\nu))} f_4(\tau) \tag{81}$$

where we define

$$f_3(\tau) = \sum_{h=0}^{H_\ell} \gamma^{h-1} \cdot \nabla_\nu r_\nu(s_h, a_h) \cdot \left( \sum_{j=0}^{H_\ell} \nabla_\theta \log \pi_{\theta^*(\nu)}(a_j | s_j) \right)^T \tag{82}$$

$$f_4(\tau) = \sum_{h=0}^{H_\ell} \gamma^{h-1} \cdot \nabla_\nu r_\nu(s_h, a_h) \cdot \left( \sum_{j=0}^{H_\ell} \nabla_\theta \log \pi_{\theta^K(\nu)}(a_j | s_j) \right)^T. \tag{83}$$

Subsequently, we write the norm of the equation (81) into 2 parts as

$$\|\nabla^2_{\nu,\theta} V_s(\nu, \theta^*(\nu)) - \nabla^2_{\nu,\theta} V_s(\nu, \theta^K(\nu))\| \leq \|\mathbb{E}_{\rho(\tau;\theta^*(\nu))} f_3(\tau) - \mathbb{E}_{\rho(\tau;\theta^K(\nu))} f_3(\tau) \tag{84}$$
$$+ \mathbb{E}_{\rho(\tau;\theta^K(\nu))} f_3(\tau) - \mathbb{E}_{\rho(\tau;\theta^K(\nu))} f_4(\tau)\|$$
$$\leq \|T_3\| + \|T_4\|,$$

where, $T_3 = \mathbb{E}_{\rho(\tau;\theta^*(\nu))} f_3(\tau) - \mathbb{E}_{\rho(\tau;\theta^K(\nu))} f_3(\tau)$ and $T_4 = \mathbb{E}_{\rho(\tau;\theta^K(\nu))} f_3(\tau) - \mathbb{E}_{\rho(\tau;\theta^K(\nu))} f_4(\tau)$. We next, upper-bound the individual terms to get the final Lispchitz constant.

First, we focus on the first term of inequality i.e $T_3$ given as

$$T_3 = \mathbb{E}_{\rho(\tau;\theta^*(\nu))} f_3(\tau) - \mathbb{E}_{\rho(\tau;\theta^K(\nu))} f_3(\tau) \tag{85}$$
$$\leq \sup_{f_3} [\mathbb{E}_{\rho(\tau;\theta^*(\nu))} f_3(\tau) - \mathbb{E}_{\rho(\tau;\theta^K(\nu))} f_1(\tau)]$$
$$\leq \chi_3 d_{TV}(\rho(\tau;\theta^*(\nu)), \rho(\tau;\theta^K(\nu)))$$
$$\leq \chi_3 \frac{H_\ell}{2} L_2 \|\theta^*(\nu) - \theta^K(\nu)\|,$$

where we convert the inequality first to a standard Integral Probability Metric form by taking the supremum. Then we divide and multiply with the norm of the function $f_3$ given by $\|f_3(\cdot)\| = \chi_3 = H_l^2 B L_r$ from equation (95), and then we get the final expression in terms of Total variation distance (from definition (53)). Then, we upper-bounded the total variation using the results from equation (70) to get the final expression. Now, we proceed to the second term of the equation $T_4$ and derive an upper bound as

$$T_4 = \sum_\tau \rho(\tau;\theta^K(\nu))(f_3(\tau) - f_4(\tau)) \tag{86}$$
$$\leq f_3(\tau') - f_4(\tau')$$
$$= \sum_{h=0}^{H_\ell} \gamma^{h-1} \cdot \nabla_\nu r_\nu(s_h, a_h) \cdot \left( \sum_{j=0}^{H_\ell} (\nabla_\theta \log \pi_{\theta^*(\nu)}(a_j | s_j) - \nabla_\theta \log \pi_{\theta^K(\nu)}(a_j | s_j)) \right)^T$$

where we consider the trajectory $\tau'$ in the sum with the maximum value and upper bound by that to get equation (86). Next, we upper-bound the norm $\|T_4\|$ as

$$\|T_4\| \leq \left\| \sum_{h=0}^{H_\ell} \gamma^{h-1} \cdot \nabla_\nu r_\nu(s_h, a_h) \cdot \left( \sum_{j=0}^{h} (\nabla_\theta \log \pi_{\theta^*(\nu)}(a_j | s_j) - \nabla_\theta \log \pi_{\theta^K(\nu)}(a_j | s_j)) \right)^T \right\|$$

$$\leq \sum_{h=0}^{H_\ell} \gamma^{h-1} \cdot \|\nabla_\nu r_\nu(s_h, a_h)\| \left( \sum_{j=0}^{h} \|\nabla_\theta \log \pi_{\theta^*(\nu)}(a_j | s_j) - \nabla_\theta \log \pi_{\theta^K(\nu)}(a_j | s_j)\| \right)$$

$$\leq \sum_{h=0}^{H_\ell} \gamma^{h-1} \cdot \|\nabla_\nu r_\nu(s_h, a_h)\| H_\ell L_1 \|\theta^*(\nu) - \theta^K(\nu)\|$$

$$\leq L_1 L_r H_\ell^2 \|\theta^*(\nu) - \theta^K(\nu)\|, \tag{87}$$

where we use Cauchy-Schwartz and triangle inequality repetitively to get to the third inequality. Next, we use Assumption 3 on the Lipschitzness of the gradient of the score function and the bounded

reward $R$ (cf. Assumption 2). Finally, we upper-bound sum of this geometric series to obtain the final expression. Adding equations (85) and (87), we get the

$$\|\nabla_{\nu,\theta}^2 V_s(\nu, \theta^*(\nu_t)) - \nabla_{\nu,\theta}^2 V_s(\nu_t, \theta^K(\nu))\| \leq L''\|\theta^*(\nu) - \theta^K(\nu)\| \tag{88}$$

where, $L'' = L_2 L_r B \frac{H_l^3}{2} + L_1 L_r H_\ell^2$. $\qquad \qquad \square$

## H  ADDITIONAL SUPPORTING LEMMAS

### H.1  PROOF OF LISPCHITZNESS AND BOUNDEDNESS CONDITION FOR $f_1(\cdot)$ DEFINED IN (74)

Here, we prove that the function denoted as $f_1(\theta^*(\nu)) = \sum_{h=0}^{H_\ell} \gamma^{h-1} \cdot r_{\nu_t}(s_h, a_h) \cdot \left(\sum_{j=0}^h \nabla_\theta^2 \log \pi_{\theta^*(\nu)}(a_j|s_j)\right)$ is Lispchitz continuous w.r.t $\theta$ with Lipschitz constant $L_{f_1}$ i.e

$$\|f_1(\theta^*(\nu)) - f_1(\theta^K(\nu))\| \leq L_{f_1}\|\theta^*(\nu) - \theta^K(\nu)\|. \tag{89}$$

First, we begin with the different term as :

$$f_1(\theta^*(\nu)) - f_1(\theta^K(\nu)) = \sum_{h=0}^{H_\ell} \gamma^{h-1} r_{\nu_t}(s_h, a_h) \left(\sum_{j=0}^h (\nabla_\theta^2 \log \pi_{\theta^*(\nu)}(a_j|s_j) - \nabla_\theta^2 \log \pi_{\theta^K(\nu)}(a_j|s_j))\right). \tag{90}$$

Subsequently, taking the norm we get

$$\begin{aligned}
\|f_1(\theta^*(\nu)) &- f_1(\theta^K(\nu))\| \\
&= \left\| \sum_{h=0}^{H_\ell} \gamma^{h-1} r_{\nu_t}(s_h, a_h) \left(\sum_{j=0}^h (\nabla_\theta^2 \log \pi_{\theta^*(\nu)}(a_j|s_j) - \nabla_\theta^2 \log \pi_{\theta^K(\nu)}(a_j|s_j))\right) \right\| \\
&\leq \sum_{h=0}^{H_\ell} \gamma^{h-1} \cdot \|r_{\nu_t}(s_h, a_h)\| \cdot \left(\sum_{j=0}^h \|\nabla_\theta^2 \log \pi_{\theta^*(\nu)}(a_j|s_j) - \nabla_\theta^2 \log \pi_{\theta^K(\nu)}(a_j|s_j)\|\right) \\
&\leq \sum_{h=0}^{H_\ell} \gamma^{h-1} \cdot \|r_{\nu_t}(s_h, a_h)\| L_2 H_\ell \|\theta^*(\nu) - \theta^K(\nu)\| \\
&\leq L_2 H_\ell^2 R\|\theta^*(\nu) - \theta^K(\nu)\|, \tag{91}
\end{aligned}$$

where we use Cauchy-Schwartz and triangle inequality to get the subsequent expressions. In the third inequality, we use the Hessian Lipschitzness assumption of score function of policy parameters from Assumption 3 and finally using the boundedness of the reward values and upper-bounding the Geometric series, we get the final expression. Thus we show that $f_1(\cdot)$ is Lispchitz continuous w.r.t $\theta$ with Lipschitz constant $L_{f_1} = L_2 H_\ell^2 R$.

Similarly, we can derive the boundedness condition on the norm of the function $\|f_1(\cdot)\|, \forall \theta$ as

$$\begin{aligned}
\|f_1\| &= \left\| \sum_{h=0}^{H_\ell} \gamma^{h-1} \cdot r_{\nu_t}(s_h, a_h) \cdot \left(\sum_{j=0}^h \nabla_\theta^2 \log \pi_\theta(a_j|s_j)\right) \right\| \\
&\leq \left\| \sum_{h=0}^{H_\ell} \gamma^{h-1} \cdot r_{\nu_t}(s_h, a_h) \right\| \cdot \left\| \left(\sum_{j=0}^h \nabla_\theta^2 \log \pi_\theta(a_j|s_j)\right) \right\| \\
&\leq H_l^2 R L_1 (= \chi_1)
\end{aligned} \tag{92}$$

where first we use Cauchy-Schwartz and triangle inequality and subsequently use the bounded reward assumption (2) and Hessian Lipschitzness assumption of the score function of the policy parameter from assumption (3).

### H.2 Proof of Lispchitzness and Boundedness for $f_3(\cdot)$ defined in (82)

Here, we prove that the function denoted as

$$f_3(\theta^*(\nu)) = \sum_{h=0}^{H_\ell} \gamma^{h-1} \cdot \nabla_\nu r_\nu(s_h, a_h) \cdot \left( \sum_{j=0}^{h} \nabla_\theta \log \pi_{\theta^*(\nu)}(a_j|s_j) \right)^T$$

is Lispchitz continuous w.r.t $\theta$ with Lipschitz constant $L_{f_3}$ i.e

$$\|f_3(\theta^*(\nu)) - f_3(\theta^K(\nu))\| \leq L_{f_3}\|\theta^*(\nu) - \theta^K(\nu)\|. \tag{93}$$

First, we begin with the difference term as :

$$f_3(\theta^*(\nu)) - f_3(\theta^K(\nu)) \tag{94}$$

$$= \sum_{h=0}^{H_\ell} \gamma^{h-1} \cdot \nabla_\nu r_\nu(s_h, a_h) \cdot \left( \sum_{j=0}^{h} (\nabla_\theta \log \pi_{\theta^*(\nu)}(a_j|s_j) - \nabla_\theta \log \pi_{\theta^*(\nu)}(a_j|s_j)) \right)^T.$$

Subsequently, taking the norm, we get

$$\|f_3(\theta^*(\nu)) - f_3(\theta^K(\nu))\|$$

$$= \left\| \sum_{h=0}^{H_\ell} \gamma^{h-1} \cdot \nabla_\nu r_\nu(s_h, a_h) \cdot \left( \sum_{j=0}^{h} (\nabla_\theta \log \pi_{\theta^*(\nu)}(a_j|s_j) - \nabla_\theta \log \pi_{\theta^*(\nu)}(a_j|s_j)) \right)^T \right\|$$

$$\leq \sum_{h=0}^{H_\ell} \gamma^{h-1} \cdot \|\nabla_\nu r_\nu(s_h, a_h)\| \cdot \left( \sum_{j=0}^{h} (\|\nabla_\theta \log \pi_{\theta^*(\nu)}(a_j|s_j) - \nabla_\theta \log \pi_{\theta^*(\nu)}(a_j|s_j)\|) \right)$$

$$\leq L_r L_1 H_\ell \|\theta^*(\nu) - \theta^K(\nu)\| \sum_{h=0}^{H_\ell} \gamma^{h-1}$$

$$\leq L_r L_1 H_\ell^2 \|\theta^*(\nu) - \theta^K(\nu)\|, \tag{95}$$

where we use Cauchy-Schwartz and triangle inequality repeatedly to get the subsequent inequalities. In the third inequality, we use the smoothness assumption of the score function of the policy parameters of Assumption 3. Finally, using the Assumption 2 i.e reward Lipschitzness and upper-bounding the Geometric series, we get the final expression. Thus $f_3(\theta^*(\nu))$ is Lispchitz continuous w.r.t $\theta$ with Lipschitz constant $L_{f_3} = L_r L_1 H_\ell^2$.

Similarly, we can derive the boundedness condition on the norm of the function $\|f_3(\cdot)\|, \forall \theta$ as

$$\|f_3\| = \left\| \sum_{h=0}^{H_\ell} \gamma^{h-1} \cdot \nabla_\nu r_\nu(s_h, a_h) \cdot \left( \sum_{j=0}^{h} \nabla_\theta \log \pi_{\theta^*(\nu)}(a_j|s_j) \right)^T \right\| \tag{96}$$

$$\leq \left\| \sum_{h=0}^{H_\ell} \gamma^{h-1} \cdot \nabla_\nu r_\nu(s_h, a_h) \right\| \cdot \left\| \left( \sum_{j=0}^{h} \nabla_\theta \log \pi_{\theta^*(\nu)}(a_j|s_j) \right) \right\|$$

$$\leq H_l^2 B L_r (= \chi_3)$$

where, first we use Cauchy-Schwartz and triangle inequality and subsequently using the Lipschitzness property in reward function assumption (2) and bounded score function of the policy parameter from assumption (3), we get the final expression.

### H.3 Proof of Smoothness Condition on the Value function

Here, we prove the smoothness of the value function i.e. the gradient of the value function is Lispchitz continuous w.r.t $\theta$ with Lipschitz constant $L_1$. We begin with the definition of the gradient difference

of the value function from the equation (15) as:

$$
\begin{aligned}
\nabla_\theta V_s(\nu, \theta^*(\nu)) - \nabla_\theta V_s(\nu_t, \theta^K(\nu)) &= \mathbb{E}_{\rho(\tau;\theta^*(\nu))} V_1(\tau) - \mathbb{E}_{\rho(\tau;\theta^K(\nu))} V_2(\tau) \\
&= \mathbb{E}_{\rho(\tau;\theta^*(\nu))} V_1(\tau) - \mathbb{E}_{\rho(\tau;\theta^K(\nu))} V_1(\tau) \\
&\quad + \mathbb{E}_{\rho(\tau;\theta^K(\nu))}[V_1(\tau) - V_2(\tau)] \\
&= \Sigma_1 + \Sigma_2,
\end{aligned}
\tag{97}
$$

where, first we substitute

$$
V_1 = \sum_{h=0}^{H_\ell - 1} \gamma^{h-1} r_{\nu_t}(s_h, a_h) \left( \sum_{j=0}^{H_\ell - 1} \nabla_\theta \log \pi_{\theta^*(\nu_t)}(a_j|s_j) \right),
\tag{98}
$$

$$
V_2 = \sum_{h=0}^{H_\ell - 1} \gamma^{h-1} r_{\nu_t}(s_h, a_h) \left( \sum_{j=0}^{H_\ell - 1} \nabla_\theta \log \pi_{\theta^K(\nu_t)}(a_j|s_j) \right).
\tag{99}
$$

Subsequently, by adding and subtracting $\mathbb{E}_{\rho(\tau;\theta^K(\nu))} V_2(\tau)$, we get the final expression, where $\Sigma_1 = \mathbb{E}_{\rho(\tau;\theta^*(\nu))} V_1(\tau) - \mathbb{E}_{\rho(\tau;\theta^K(\nu))} V_1(\tau)$ and $\Sigma_2 = \mathbb{E}_{\rho(\tau;\theta^K(\nu))}[V_1(\tau) - V_2(\tau)]$. Now, first, we derive the Lipschitz constant for $V_1$ as

$$
\begin{aligned}
\|V_1(\theta^*(\nu) &- V_1(\theta^K(\nu))\| \\
&\leq \sum_{h=0}^{H_\ell - 1} \gamma^{h-1} \|r_{\nu_t}(s_h, a_h)\| \left( \sum_{j=0}^{H_\ell - 1} \|\nabla_\theta \log \pi_{\theta^*(\nu_t)}(a_j|s_j) - \nabla_\theta \log \pi_{\theta^K(\nu_t)}(a_j|s_j)\| \right) \\
&\leq H_l^2 R L_1 \|\theta^*(\nu) - \theta^K(\nu)\|,
\end{aligned}
\tag{100}
$$

where we first use Cauchy-Schwartz and triangle inequality to get the first inequality. Next, we upper-bound the reward with $R$ from Assumption 2, Lipschitzness of policy gradient from Assumption 3, and finally upper-bounding the Geometric series, we get the final expression. The Lipschitz constant $L_5 = H_l^2 R L_1$.

We can subsequently upper-bound $\Sigma_1$ with the total variation distance as

$$
\begin{aligned}
\Sigma_1 &\leq \sup_V [\mathbb{E}_{\rho(\tau;\theta^*(\nu))} V(\tau) - \mathbb{E}_{\rho(\tau;\theta^K(\nu))} V(\tau)] \\
&\leq L_5 d_{TV}(\rho(\tau;\theta^K(\nu)), \rho(\tau;\theta^*(\nu))) \\
&\leq L_5 \frac{HL_2}{2} \|\theta^*(\nu) - \theta^K(\nu)\|
\end{aligned}
\tag{101}
$$

where first we divide by the Lipschitz constant of the function and subsequently upper-bound with the Total Variation distance. Finally, we substitute the total variation distance expression from the equation (70) to get the final expression. Now, the second term can be written as

$$
\Sigma_2 = \mathbb{E}_{\rho(\tau;\theta^K(\nu))}[V_1(\tau) - V_2(\tau)] \leq V_1(\tau') - V_2(\tau'),
\tag{102}
$$

where we take the trajectory with the maximum difference and upper-bound the term. Subsequently, $\|\Sigma_2\| \leq \|V_1(\tau') - V_2(\tau')\| = H_l^2 R L_{\theta_1} \|\theta^*(\nu) - \theta^K(\nu)\|$ from equation (100).

Finally, the norm of the gradient difference of the value function from equation (97) as

$$
\|\nabla_\theta V_s(\nu, \theta^*(\nu)) - \nabla_\theta V_s(\nu_t, \theta^K(\nu))\| \leq L_6 \|\theta^*(\nu) - \theta^K(\nu)\|
\tag{103}
$$

where $L_6 = L_5 \frac{HL_2}{2} + L_5$ and $L_5 = H_l^2 R L_{\theta_1}$

### H.4 UPPER-BOUND ON THE NORM OF THE HESSIAN FOR THE BILEVEL RL

Here, we prove an upper-bound on the norm of the hessian defined in (16) given as

$$\nabla_\theta^2 V_s(\nu_t, \theta^K(\nu_t)) = \mathbb{E}_{\rho(\tau;\theta^K(\nu_t))}\left[\sum_{h=0}^{H_\ell}\gamma^{h-1}r_{\nu_t}(s_h,a_h)\left(\sum_{j=0}^{h}\nabla_\theta^2\log\pi_{\theta^K(\nu_t)}(a_j|s_j)\right)\right] \quad (104)$$

$$\leq \sum_{h=0}^{H_\ell}\gamma^{h-1}r_{\nu_t}(s_h,a_h)\left(\sum_{j=0}^{h}\nabla_\theta^2\log\pi_{\theta^K(\nu_t)}(a_j|s_j)\right)\sum_\tau\rho(\tau;\theta^K(\nu_t))$$

$$= \sum_{h=0}^{H_\ell}\gamma^{h-1}r_{\nu_t}(s_h,a_h)\left(\sum_{j=0}^{H_\ell}\nabla_\theta^2\log\pi_{\theta^K(\nu_t)}(a_j|s_j)\right),$$

where, first we upper-bound the expression by considering the the trajectory which has the maximum lower-value and $\sum_\tau\rho(\tau;\theta^K(\nu_t))=1$. Next, we, upper-bound the norm as

$$\|\nabla_\theta^2 V_s(\nu_t,\theta^K(\nu_t))\| \leq \|\sum_{h=0}^{H_\ell}\gamma^{h-1}r_{\nu_t}(s_h,a_h)\left(\sum_{j=0}^{h}\nabla_\theta^2\log\pi_{\theta^K(\nu_t)}(a_j|s_j)\right)\| \quad (105)$$

$$\leq \sum_{h=0}^{H_\ell}\gamma^{h-1}\|r_{\nu_t}(s_h,a_h)\|\left(\sum_{j=0}^{h}\|\nabla_\theta^2\log\pi_{\theta^K(\nu_t)}(a_j|s_j)\|\right)$$

$$\leq H_l^2 R L_\pi^1,$$

where, we first upper-bound with successive application of Cauchy-Schwartz and Triangle inequality to get the second inequality. Finally, with the upper bound on $\|r_{\nu_t}(s_h,a_h)\| \leq R$ from Assumption (2) and smoothness condition on the policy gradients from Assumption 3 and upper-bounding the geometric series, we get the final expression.

### H.5 UPPER-BOUND ON THE NORM OF THE 2ND ORDER MIXED-JACCOBIAN TERM FOR THE BILEVEL RL

Here, we prove an upper-bound on the norm of the mixed second-order Jacobian term defined in (17) given as

$$\nabla_{\nu,\theta}^2 V_s(\nu_t,\theta^K(\nu_t)) = \mathbb{E}_{P(\tau;\theta^K(\nu_t))}\left[\sum_{h=0}^{H_\ell}\gamma^{h-1}\nabla_\nu r_{\nu_t}(s_h,a_h)\left(\sum_{j=0}^{h}[\nabla_\theta\log\pi_{\theta^K(\nu_t)}(a_j|s_j)]^T\right)\right]$$

$$\leq \sum_{h=0}^{H_\ell}\gamma^{h-1}\nabla_\nu r_{\nu_t}(s_h,a_h)\left(\sum_{j=0}^{h}[\nabla_\theta\log\pi_{\theta^K(\nu_t)}(a_j|s_j)]^T\right), \quad (106)$$

where first we upper-bound the expression with the trajectory which has the maximum lower value. Next, we, upper-bound the norm as

$$\|\nabla_{\nu,\theta}^2 V_s(\nu_t,\theta^K(\nu_t))\| \leq \|\sum_{h=0}^{H_\ell}\gamma^{h-1}\nabla_\nu r_{\nu_t}(s_h,a_h)\left(\sum_{j=0}^{h}[\nabla_\theta\log\pi_{\theta^K(\nu_t)}(a_j|s_j)]^T\right)\| \quad (107)$$

$$\leq \sum_{h=0}^{h}\gamma^{h-1}\|\nabla_\nu r_{\nu_t}(s_h,a_h)\|\left(\sum_{j=0}^{h}\|\nabla_\theta\log\pi_{\theta^K(\nu_t)}(a_j|s_j)\|\right)$$

$$\leq H_l^2 L_r B,$$

where we first upper-bound with the successive application of Cauchy-Schwartz and Triangle inequality to get the second inequality. Finally, with the upper bound on $\|\nabla_\nu r_{\nu_t}(s_h,a_h)\| \leq L_r$ from smoothness condition on the reward function from Assumption (2), bounded score function of policy parameter from Assumption 3 and upper-bounding the geometric series, we get the final expression. We denote $L_{\nu,\theta} = H_l^2 L_r B$ for simplicity.

## H.6 Proof of Lemma 4

*Proof.* Let us start by first deriving the upper bounds for the terms $\phi_1$ and $\phi_2$ as defined in the equation (54) as follows. For $\phi_1(\tau)$, we have

$$\|\phi_1(\tau)\| = \left\| U_\nu(\tau) \cdot \sum_{h=0}^{H-1} [-\nabla_{v,\theta}^2 V_s(\nu, \theta^*(\nu))\nabla_\theta^2 V_s(\nu, \theta^*(\nu))^{-1}\nabla_\theta f_h(\theta^*(\nu))] + \nabla_\nu U_\nu(\tau) \right\|. \tag{108}$$

We define the term $\kappa$ which explains the relative conditioning of the two matrix norms. We define the mixed condition number as

$$\kappa = \frac{\|\nabla_{v,\theta}^2 V_s(\nu, \theta^*(\nu))\|}{\|\nabla_\theta^2 V_s(\nu, \theta^*(\nu))\|} \leq \frac{H_l L_r B}{l_\pi(1-\gamma)} \tag{109}$$

Next, to upper-bound the second term relating to the difference in $\|\phi_1 - \phi_2\|$ in the equation, we proceed first by upper-bounding the product difference

$$\|\Delta\| = \|\Delta_1 - \Delta_2\|, \tag{110}$$

where we define

$$\Delta_1 = \nabla_{v,\theta}^2 V_s(\nu, \theta^*(\nu))\nabla_\theta^2 V_s(\nu, \theta^*(\nu))^{-1}\nabla_\theta f_h(\theta^*(\nu)), \tag{111}$$

$$\Delta_2 = \nabla_{v,\theta}^2 V_s(\nu, \theta^K(\nu))\nabla_\theta^2 V_s(\nu, \theta^K(\nu))^{-1}\nabla_\theta f_h(\theta^K(\nu)). \tag{112}$$

Also, for simplicity of notation, let us take

$$\psi_1 = \nabla_{v,\theta}^2 V_s(\nu, \theta^*(\nu))\nabla_\theta^2 V_s(\nu, \theta^*(\nu))^{-1}, \tag{113}$$

$$\text{and } \psi_2 = \nabla_{v,\theta}^2 V_s(\nu, \theta^K(\nu))\nabla_\theta^2 V_s(\nu, \theta^K(\nu))^{-1}, \tag{114}$$

which thus (110) boils down to upper-bounding

$$\begin{aligned}
\Delta &= \|\psi_1 \nabla_\theta f_h(\theta^*(\nu)) - \psi_2 \nabla_\theta f_h(\theta^K(\nu))\| \\
&= \|\psi_1 \nabla_\theta f_h(\theta^*(\nu)) - \psi_1 \nabla_\theta f_h(\theta^K(\nu)) + \psi_1 \nabla_\theta f_h(\theta^K(\nu)) - \psi_2 \nabla_\theta f_h(\theta^K(\nu))\| \\
&\leq \|\psi_1\|\|\nabla_\theta f_h(\theta^*(\nu)) - \nabla_\theta f_h(\theta^K(\nu))\| + \|\psi_1 - \psi_2\|\|\nabla_\theta f_h(\theta^K(\nu))\| \\
&\leq \kappa L_1 \|\theta^*(\nu) - \theta^K(\nu)\| + B\|\psi_1 - \psi_2\|,
\end{aligned} \tag{115}$$

where, first we expand add and subtract the term $\psi_1 \nabla_\theta f_h(\theta^K(\nu))$, and subsequently by applying Cauchy-Schwartz and triangle inequality, we get to the third inequality. For the final inequality, we apply equation and Lispchitzness assumptions on the score function of the policy parameter Assumption (3) to get the final expression in equation (115). Next, we focus on upper bounding the second term of the expression specifically $\|\psi_1 - \psi_2\|$

$$\begin{aligned}
\|\psi_1 - \psi_2\| &= \|\nabla_{v,\theta}^2 V_s(\nu, \theta^*(\nu))\nabla_\theta^2 V_s(\nu, \theta^*(\nu))^{-1} - \nabla_{v,\theta}^2 V_s(\nu, \theta^*(\nu))\nabla_\theta^2 V_s(\nu, \theta^K(\nu))^{-1} \\
&\quad + \nabla_{v,\theta}^2 V_s(\nu, \theta^*(\nu))\nabla_\theta^2 V_s(\nu, \theta^K(\nu))^{-1} - \nabla_{\nu,\theta}^2 V_s(\nu, \theta^K(\nu))\nabla_\theta^2 V_s(\nu, \theta^K(\nu))^{-1}\| \\
&= \|\nabla_{\nu,\theta}^2 V_s(\nu, \theta^*(\nu))\nabla_\theta^2 V_s(\nu, \theta^*(\nu))^{-1} - \nabla_{v,\theta}^2 V_s(\nu, \theta^*(\nu))\nabla_\theta^2 V_s(\nu, \theta^K(\nu))^{-1}\| \\
&\quad + \|\nabla_{\nu,\theta}^2 V_s(\nu, \theta^*(\nu))\nabla_\theta^2 V_s(\nu, \theta^K(\nu))^{-1} - \nabla_{v,\theta}^2 V_s(\nu, \theta^K(\nu))\nabla_\theta^2 V_s(\nu, \theta^K(\nu))^{-1}\| \\
&= \Psi_{21} + \Psi_{22},
\end{aligned} \tag{116}$$

where we expand the definition of $\|\psi_1 - \psi_2\|$ and subsequently apply triangle inequality and Cauchy-Schwartz which then boils to upper-bounding the sum of two terms $\Psi_{21} + \Psi_{22}$. For $\Psi_{21}$, we have

$$\begin{aligned}
\Psi_{21} &\leq \|\nabla_{\nu,\theta}^2 V_s(\nu, \theta^*(\nu))\|\|\nabla_\theta^2 V_s(\nu, \theta^*(\nu))^{-1} - \nabla_{v,\theta}^2 V_s(\theta^K(\nu))^{-1}\| \tag{117} \\
&\leq L_{\nu,\theta}\|\nabla_\theta^2 V_s(\nu, \theta^*(\nu))^{-1} - \nabla_\theta^2 V_s(\nu, \theta^K(\nu))^{-1}\| \\
&= L_{\nu,\theta}\|\nabla_\theta^2 V_s(\nu, \theta^*(\nu))^{-1}(\nabla_\theta^2 V_s(\nu, \theta^*(\nu)) - \nabla_\theta^2 V_s(\nu, \theta^K(\nu)))\nabla_\theta^2 V_s(\nu, \theta^K(\nu))^{-1}\| \\
&\leq L_{\nu,\theta}\|\nabla_\theta^2 V_s(\nu, \theta^*(\nu))\|^{-1}\|(\nabla_\theta^2 V_s(\nu, \theta^*(\nu)) - \nabla_\theta^2 V_s(\nu, \theta^K(\nu)))\|\|\nabla_\theta^2 V_s(\nu, \theta^K(\nu))\|^{-1} \\
&\leq \frac{L_{\nu,\theta}L'}{l_\pi^2}\|\theta^*(\nu) - \theta^K(\nu)\|,
\end{aligned}$$

where we use Cauchy-Schwartz inequality and triangle inequality iteratively to get the final inequality. Next, we use the upper bounds and lower bounds of the hessian and mixed hessian matrices defined in Assumptions to get the final expression. Now, $L_{\nu,\theta} = H_l^2 L_r B$ from equation (107), $L' = L_{f_1}\chi_1\frac{H_\ell}{2}L_2 + L_2 R H_\ell^2$ and $L_{f_1} = L_2 H_\ell^2 R$ from equation (80). Finally, the second-term $\Psi_{22}$ from equation (116) can be upper-bounded as

$$
\begin{aligned}
\Psi_{22} &\leq \|\nabla_{\nu,\theta}^2 V_s(\nu,\theta^*(\nu) - \nabla_{v,\theta}^2 V_s(\nu,\theta^K(\nu))\|\|\nabla_\theta^2 V_s(\nu,\theta^K(\nu))\|^{-1} \\
&\leq \frac{L''}{l_\pi}\|\theta^*(\nu) - \theta^K(\nu)\|,
\end{aligned}
\tag{118}
$$

where, similarly we use triangle inequality with Cauchy-Schwartz to get the final upper-bound of equation (118). Now, $L'' = L_{f_3}\chi_2\frac{H_\ell}{2}L_2 + L_1 L_r H_\ell^2$ and $L_{f_3} = L_r L_1 H_\ell^2$ from equation (88).

Now, combining equations (117) and (118), we get the final upper-bound of the $\|\psi_1 - \psi_2\|$ in equation (116) as

$$
\|\psi_1 - \psi_2\| \leq \left(\frac{L_{\nu,\theta}L'}{l_\pi^2} + \frac{L''}{l_\pi}\right)\|\theta^*(\nu) - \theta^K(\nu)\|.
\tag{119}
$$

Hence, finally replacing the upper-bound of $\Psi_2$ from equation (119) in equation (115) to obtain the upper-bound on the function difference term $\Delta$ as

$$
\begin{aligned}
\Delta &\leq \kappa L_1\|\theta^*(\nu) - \theta^K(\nu)\| + \left(\frac{L_{\nu,\theta}L'}{l_\pi^2} + \frac{L''}{l_\pi}\right)\|\theta^*(\nu) - \theta^K(\nu)\| \\
&= \gamma_1\|\theta^*(\nu) - \theta^K(\nu)\|,
\end{aligned}
\tag{120}
$$

with $\gamma_1 := \kappa L_1 + \frac{L_{\nu,\theta}L'}{l_\pi^2} + \frac{L''}{l_\pi}$ Hence, with the above bounds, we proceed to upper-bound the term II in equation (51) i.e $\|\mathbb{E}_{\tau\sim\rho(\tau;\theta^K(\nu))}[\phi_1(\tau) - \phi_2(\tau)]\|$ as

$$
\begin{aligned}
\|\mathbb{E}_{\tau\sim\rho(\tau;\theta^K(\nu))}[\phi_1(\tau) - \phi_2(\tau)]\| &\leq \|\phi_1(\tau') - \phi_1(\tau')\| \\
&= \|U(\tau)\cdot\sum_{h=0}^{H-1}(\Delta_1 - \Delta_2)\| \\
&\leq \|\sum_{h'=0}^{H-1}u(s_h,a_h)\|\|\sum_{h=0}^{H-1}(\Delta_1 - \Delta_2)\| \\
&\leq H^2\tilde{u}\|\Delta_1 - \Delta_2\| \\
&\leq H^2\tilde{u}\gamma_1\|\theta^*(\nu) - \theta^K(\nu)\|,
\end{aligned}
\tag{121}
$$

where first we select the trajectory $\tau'$ with the maximum sum and subsequently using the Cauchy-Schwartz inequality we get the second equation. Based on the assumption of bounded utility $u(s,a) \leq \tilde{u}, \forall(s,a)$, we get the third equation and finally using the upper bound of $\Delta_1 - \Delta_2$ from equation (120), we get the final expression for equation (121). $\qquad\square$

### H.7 PROOF OF LEMMA 2

*Proof.* Here, we derive an upper bound on the tracking term due to the use of surrogate gradients $\|\theta^*(\nu_t) - \theta^K(\nu_t)\|$. To begin the proof, we start with the smoothness in the value function shown in equation (103), as

$$
\begin{aligned}
V_s(\nu_t,\theta^{k+1}(\nu_t)) &\leq V_s(\nu_t,\theta^k(\nu_t)) + \langle\nabla_\theta V_s(\nu_t,\theta^{k+1}(\nu_t)),\theta^{k+1}(\nu_t) - \theta^k(\nu_t)\rangle \\
&\quad + \frac{L_6}{2}\|\theta^{k+1}(\nu_t) - \theta^k(\nu_t)\|^2.
\end{aligned}
\tag{122}
$$

where $L_6 = L_5\frac{H_\ell^2 L_2}{2} + L_5$ and $L_5 = H_l R L_{\theta_1}$. Now, from the update of $\theta$ from Algorithm 2, we know that

$$
\theta^{k+1}(\nu_t) = \theta^k(\nu_t) - \alpha_\ell\nabla_\theta V_s(\nu_t,\theta^k(\nu_t)).
\tag{123}
$$

Replacing the update in equation (122), we have

$$
\begin{aligned}
V_s(\nu_t, \theta^{k+1}(\nu_t)) \leq & V_s(\nu_t, \theta^k(\nu_t)) + \langle \nabla_\theta V_s(\nu_t, \theta^{k+1}(\nu_t)), -\alpha_\ell \nabla_\theta V_s(\nu_t, \theta^k(\nu_t)) \rangle \\
& + \frac{L_6}{2} \| -\alpha_\ell \nabla_\theta V_s(\nu_t, \theta^k(\nu_t)) \|^2 \\
= & V_s(\nu_t, \theta^k(\nu_t)) - \alpha_\ell \left(1 - \frac{\alpha_\ell L_6}{2}\right) \|\nabla_\theta V_s(\nu_t, \theta^k(\nu_t))\|^2,
\end{aligned} \tag{124}
$$

where we expand the expression after replacing the update in equation (122). Next, from Assumption 4, we note that for the value function, it holds that

$$
\|\nabla_\theta V_s(\nu, \theta^k(\nu))\|^2 \geq \frac{\mu}{2}(V_s(\nu, \theta^k(\nu)) - V_s(\nu, \theta^*(\nu))). \tag{125}
$$

Assumption 4 ensures that the objective satisfies the gradient dominance or the PL condition, which can be satisfied in practice for various settings. For instance, Assumption 4 can be satisfied in our setting for softmax policy parametrization Now, replacing the PL condition in equation (124), we have

$$
\begin{aligned}
V_s(\nu_t, \theta^{k+1}(\nu_t)) - V_s(\nu_t, \theta^k(\nu_t)) & \leq -\alpha_\ell(1 - \frac{\alpha_\ell L_6}{2})\frac{\mu}{2}(V_s(\nu, \theta^k(\nu)) - V_s(\nu, \theta^*(\nu))) \\
& = -\alpha_3(V_s(\nu_t, \theta^k(\nu_t)) - V_s(\nu_t, \theta^*(\nu_t))),
\end{aligned} \tag{126}
$$

where, after replacing the PL condition in equation (124), we substitute $\alpha_3 = \alpha_\ell(1 - \frac{\alpha_\ell L_6}{2})\frac{\mu}{2}$ for simplicity of calculations.

$$
\begin{aligned}
V_s(\nu_t, \theta^{k+1}(\nu_t)) - V_s(\nu_t, \theta^*(\nu_t))) & \leq (1 - \alpha_3)(V_s(\nu_t, \theta^k(\nu_t)) - V_s(\nu_t, \theta^*(\nu_t))) \\
V_s(\nu_t, \theta^K(\nu_t)) - V_s(\nu_t, \theta^*(\nu_t))) & \leq (1 - \alpha_3)^K(V_s(\nu_t, \theta^0(\nu_t)) - V_s(\nu_t, \theta^*(\nu_t))),
\end{aligned} \tag{127}
$$

where the first equation comes from algebraic manipulation and applying the equation recursively, we get the second inequality, assuming $0 \leq \alpha_3 \leq 1$. Now, we note that from the smoothness of value function, we have the upper bound

$$
V_s(\nu_t, \theta^0(\nu_t)) - V_s(\nu_t, \theta^*(\nu_t)) \leq \frac{L_6}{2}\|\theta^*(\nu_t) - \theta^0(\nu_t)\|^2, \tag{128}
$$

where $L_6 = L_5\frac{HL_2}{2} + L_5$, $L_5 = H_l^2 R L_1$ and we use the Lipschitz smoothness assumption and expand along the point $\nu_t, \theta^*$, for which the gradient term vanishes. Also, since PL implies quadratic growth, it holds that

$$
V_s(\nu_t, \theta^K(\nu_t)) - V_s(\nu_t, \theta^*(\nu_t)) \geq \frac{\mu}{2}\|\theta^K(\nu_t) - \theta^*(\nu_t)\|^2. \tag{129}
$$

Now, substituting the equations (128), (129) in (127) to obtain

$$
\|\theta^K(\nu_t) - \theta^*(\nu)\|^2 \leq (1 - \alpha_3)^K\frac{L_6}{\mu}Z, \tag{130}
$$

where, $Z := \max_\nu \|\theta^0 - \theta^*(\nu)\|^2$, $\alpha_3 = \alpha_\ell(1 - \frac{\alpha_\ell L_6}{2})\frac{\mu}{2}$ and $L_6 = L_5\frac{H_\ell L_2}{2} + L_5$ and $L_5 = H_l^2 R L_1$. $\qquad \square$

# I    Discuss of Assumption 3 and Assumption 4

- Assumption 3 ensure certain properties of the policy parametrization such as Lipschitz policy, bounded score function, Lipschitz score function, and Lipschitz Hessian of the log of the policy. We remark that these assumptions are not restrictive and satisfied in practice for practical classes of policies. For example, this assumption is satisfied for softmax policy parametrization.

$$
\pi_\theta(a|s) = \frac{\exp \theta^T \phi(s, a)}{\sum_{a'} \exp \theta^T \phi(s, a')}, \tag{131}
$$

where, first we write down the expression of the softmax policy gradient parametrization with function approximation $\phi$. Subsequently, taking log and taking the gradient w.r.t to the policy parameterization using chain-rule , we get

$$\nabla_\theta \log \pi_\theta(a|s) = \phi(s,a) - \frac{1}{\sum_{a'} \exp(\theta^T \phi(s,a'))} \sum_{a'} \exp(\theta^T \phi(s,a'))\phi(s,a') \qquad (132)$$

$$= \phi(s,a) - \sum_{a'} \frac{\exp(\theta^T \phi(s,a'))}{\sum_{a''} \exp(\theta^T \phi(s,a''))}\phi(s,a')$$

$$= \phi(s,a) - \sum_{a'} \pi_\theta(a'|s)\phi(s,a')$$

$$= \phi(s,a) - \widehat{\phi}_s$$

where we can denote $\widehat{\phi}_s = \sum_{a'} \pi_\theta(a'|s)\phi(s,a')$. Now, taking the norm on the LHS of equation (132), we get

$$\|\nabla_\theta \log \pi_\theta(a|s)\| \leq \|\phi(s,a)\| + \|\phi(s,a')\| \qquad (133)$$
$$= 2\zeta_1$$

where we apply triangle inequality to get the final bound, which imposes certain constraints on the norm of the function approximation specifically $\|\phi(s,a)\| \leq \zeta_1$ which is a common assumption in various scenarios (Sutton et al., 2009; Maei et al., 2009). Thus this shows how for soft-max policy with function approximations satisfy Assumptions (3). where, we expand $\nabla_\theta \log \pi_\theta(a|s) = \frac{1}{\pi_\theta(a|s)}\nabla_\theta \pi_\theta(a|s)$.

- Assumption 4 ensures that the objective function satisfies certain geometric properties such as PL condition. We remark that the value function satisfies PL condition with softmax policy parametrization (see Lemma8 (Mei et al., 2020b)). Further, a property that we need for our analysis to hold is that the Hessian of the objective function has all non-zero eigenvalues. This assumption holds for softmax parametrizations. Now, in order to show that we first consider the softmax-parametrization considered in equation (132), we first compute the hessian as

$$\nabla_\theta^2 \log \pi_\theta(a|s) = \nabla_\theta[\phi(s,a) - \sum_{a'} \pi_\theta(a'|s)\phi(s,a')] \qquad (134)$$

$$= -\sum_{a'} \nabla_\theta \pi_\theta(a'|s)\phi(s,a')^T$$

$$= -\mathbb{E}_\pi[\nabla_\theta \log \pi_\theta(a'|s)\phi(s,a')^T]$$

$$= \mathbb{E}_\pi[\widehat{\phi}(s)\phi(s,a')^T - \phi(s,a')\phi(s,a')^T]$$

Now, finally, we substitute this to the equation of hessian of the value function in equation (134)

$$\nabla_\theta^2 V_s(\nu_t, \theta^K(\nu_t)) = \mathbb{E}_{\rho(\tau;\theta)}\left[R(\tau)\left(\sum_{j=0}^{H_\ell-1} \nabla_\theta^2 \log \pi_{\theta^K(\nu_t)}(a_j|s_j)\right)\right] \qquad (135)$$

$$= \mathbb{E}_{\rho(\tau;\theta)}\mathbb{E}_\pi\left[R(\tau)\left(\sum_{j=0}^{H_\ell-1}[\widehat{\phi}(s)\phi(s,a')^T - \phi(s,a')\phi(s,a')^T]\right)\right]$$

From the above, it is evident to ensure non-singular eigenvalues, we need to ensure non-singularity for the function approximation matrix $\phi\phi^T$ which has been a standard assumption in several settings (Sutton et al., 2009; Maei et al., 2009).

## J  ADDITIONAL EXPERIMENTAL DETAILS

**Implementation details of our Algorithm** For the experiments, we use the same configuration of hyperparameters for all the baselines (Lee et al., 2021; Park et al., 2022) including learning rate, optimizer, etc. PEBBLE with SURF improves the performance of PEBBLE significantly as also shown in (Park et al., 2022). However, it depends on the quality and amount of augmentations used

| Hyperparameter | Value | Hyperparameter | Value |
|---|---|---|---|
| Initial temperature | 0.1 | Hidden units per each layer | 1024 (DMControl), 256 (Meta-world) |
| Learning rate | 0.001 | Batch Size | 1024 (DMControl), 512 (Meta-world) |
|  | 0.0005 (walker) | Optimizer | Adam |
| Critic target update freq | 2 | Critic EMA $\tau$ | 0.005 |
| $(\beta_1, \beta_2)$ | $(.9, .999)$ | Discount $\gamma$ | .99 |

Table 2: Hyperparameters of the SAC algorithm. We use similar hyperparameter configurations as in (Lee et al., 2021; Park et al., 2022)

and the performance of PEBBLE+SURF varies rapidly. We used a limited amount of augmentations and pseudo-labels for a fair comparison. However, increasing the quality and amount significantly can improve the performance of PEBBLE+SURF. Hence, naturally leveraging augmentations in our unified framework with PARL will be an interesting direction of future research. We observed the effect of augmentation much more significant in DM-Control Suite than in MetaWorld. For the implementation, we have leveraged standard repositories like Torchopt (or BOML in Tensorflow) which are widely used to evaluate the meta gradient. We have also leveraged first-order approximate algorithms for performing and replicating our results, which improves the computation traceability. A rigorous and thorough evaluation of multiple other environments remains a scope for future study.

## K    ADDITIONAL EXPERIMENTAL DESCRIPTIONS

In this section, we describe the environmental setup and configurations leveraged in the experiment with more specific details. To demonstrate the performance of our algorithm we primarily leverage 4 environments (two Locomotion and two manipulation) from DM-control Suite ((Tassa et al., 2018) and Meta World (Yu et al., 2021) in the human preference-based RL setting as also in (Christiano et al., 2017; Lee et al., 2021; Park et al., 2022). First, we describe the environments for the same

- **Walker (Tassa et al., 2018)** This is a bipedal walker introduced in (Lillicrap et al., 2015) and later leveraged with some modifications in (Tassa et al., 2018) where the objective of the agent is to move forward as quickly as possible without falling down or pitching the torso too far forward or backward and the reward is a combination of the above.

- **Cheetah (Tassa et al., 2018)** Cheetah or Half-Cheetah as introduced by (Wawrzyński, 2009) is a 6 degrees of freedom planar robot composed of nine links, eight joints, and two paws where angles of the fourth and the fifth joint are fixed while others are controllable. The objective is to make the cheetah run as fast as possible and the reward is directly proportional to its forward velocity but only up to a speed threshold. Later, this was leveraged in the (Tassa et al., 2018) with minor modifications.

- **Door Open and Button Press (Yu et al., 2021)**: These are taken from the (Yu et al., 2021) suite of 50 diverse simulated manipulation tasks all of which are contained in a shared, table-top environment with a simulated Sawyer arm. For the Door open environment, the objective of the agent is to control the robotic arm to open a door with a revolving joint for randomized door positions. Similarly, for Button press, the objective of the agent is to control the arm to press a button for randomized button positions. button is random

**Design Simulated Human Feedback.** Although it would be ideal to evaluate the real-world effectiveness of our algorithm based on actual human feedback, for simulation and comparisons on benchmark it is hard to collect a large amount of human feedback. Hence, to emulate human feedback, we leverage simulated human teachers as used in prior research (Lee et al., 2021; Park et al., 2022), whose preferences are based on ground-truth reward functions which help us to evaluate the agent efficiently. Next, we discuss simulating human preferences using the ground-truth reward model as in (Lee et al., 2021).

Let's say $r^*(s, a)$ represents the ground-truth reward function for state-action $s, a$ in the environments as discussed above based on the objectives, which are known to us. Now, the perfectly rational and

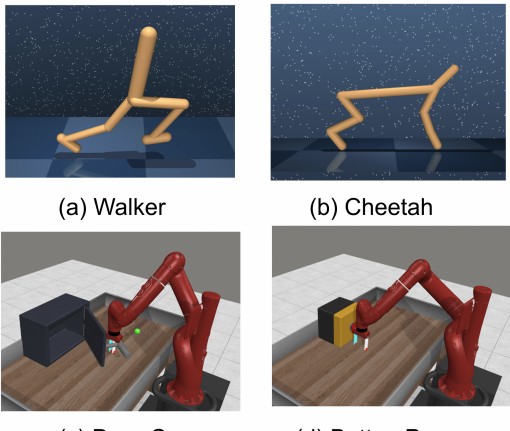

(a) Walker     (b) Cheetah

(c) Door Open     (d) Button Press

Figure 5: Figure demonstrates the visual representation of the environments considered in the experimental setup

deterministic human behavior would result in trajectory preferences $(\tau_0, \tau_1)$ as

$$
y = \begin{cases} (1,0) & \text{If } \sum_{t=1}^{H} r^*(s_t^0, a_t^0) > \sum_{t=1}^{H} r^*(s_t^1, a_t^1) \\ (0,1) & \text{otherwise,} \end{cases} \tag{136}
$$

where, superscript $0, 1$ represents the state-action in the trajectory $\tau_0, \tau_1$ respectively. However, human preferences are not always deterministic or rational as discussed in (Lee et al., 2021). Hence, the stochastic preference model can be written as

$$
P[\tau_0 > \tau_1; \beta, \gamma] = \frac{\exp\left(\beta \sum_{t=1}^{H} \gamma^{H-t} r(s_t^0, a_t^0)\right)}{\exp\left(\beta \sum_{t=1}^{H} \gamma^{H-t} r(s_t^0, a_t^0)\right) + \exp\left(\beta \sum_{t=1}^{H} \gamma^{H-t} r(s_t^1, a_t^1)\right)} \tag{137}
$$

where the preference model is the well-known Bradley Terry model (Bradley & Terry, 1952) $\gamma \in [0, 1)$ is the discount factor and $\beta$ is the rationality constant. For example, $\beta \to \infty$ indicates a perfectly rational human, and $\beta = 0$ indicates a noisy/sub-optimal human producing random choices. To design further human-like teachers, various human behaviors like myopic behavior, mistakes, etc. are integrated while generating preferences as in Lee et al. (2021); Park et al. (2022).

Now, as the preference-generating mechanism becomes clear, it is evident that the oracle reward in the plots indicates the ground truth $r^*$ generating the preferences. The upper-level objective deals with maximizing the likelihood of the human preferences received on the trajectories generated by the inner optimal policy under the current reward estimate. In all the environments, A-PARL achieves near-oracle performance in a much faster time. This highlights the importance of our bi-level framework, which considers the dependence (missing from existing literature) of distribution on the lower-level policy parameter during training the upper-level objective.

