# OpenReview forum: "PARL: A Unified Framework for Policy Alignment in Reinforcement Learning from Human Feedback"
_ICLR.cc/2024/Conference — ICLR 2024 poster_

### Official Review · Reviewer_fXDT · 2023-10-31

**Soundness:** 3 good
**Presentation:** 3 good
**Contribution:** 3 good
**Rating:** 8
**Confidence:** 3

**Summary:**

This paper studies the policy alignment in reinforcement learning and proposes a unified framework, PARL, which is a new bilevel optimization framework to address existing problems in the previous works. Specifically, the PARL explicitly considers the dependence of the alignment objective on the data generated by policy trajectories (the alignment goal is evaluated on the learned policy, which also generates the data). In the algorithmic side, the authors propose A-PARL, which is provably efficient, to solve the problems under the PARL framework and empirical results are provided.

**Strengths:**

1 The paper first identifies a good point on the existing problem of the previous works. The argument that the alignment objective in the upper-level depends on the lower level through the learned policy is convincing and is clearly clarified. These arguments make the proposed PARL framework natural and intuitive. This is the main point I would like to recommend acceptance for the work.

2 The paper further proposes an algorithm named A-PARL to solve the PARL problem, which is with sound mathematical guarantee under suitable assumptions and demonstrate impressive empirical performances. Based on the literature review, I think the baselines in the experiment part are competitive, which further support the advantage of the proposed framework and algorithm.

3 The writing is good and the paper is easy to follow.

**Weaknesses:**

1 I am not very familiar with the literature of the benchmarks used in this area but I go through the papers along the line and confirm that the experiments conducted by the authors are commonly adopted in the literature so I assume that the experiments are satisfactory (which may need to be further evaluated by other expert reviewers).

2 Could you comment on the generality of the proposed framework? In particular, I am interested in whether this framework can be extended to handle the alignment of language model because the reward model quality is the bottleneck in many cases and most of the existing approaches use only a fixed reward model.

**Questions:**

see weakness

---

> ### Author Response · Authors · 2023-11-14
> **Response to Reviewer fXDT**
>
> **General Response:** Thank you for your comprehensive review and positive feedback. We are pleased that our PARL framework approach to policy alignment and the effectiveness of the A-PARL algorithm resonated with you. Your recognition of the clarity of our work and the rigor of our experimental comparisons is greatly appreciated. We provide a detailed response to your comments as follows.
>
>
> ## Weakness
>
> > Weakness 1 :  I am not very familiar with the literature of the benchmarks used in this area but I go through the papers along the line and confirm that the experiments conducted by the authors are commonly adopted in the literature so I assume that the experiments are satisfactory (which may need to be further evaluated by other expert reviewers).
>
>
> **Response to Weakness 1:** Thank you for your feedback regarding the benchmarks used in our experiments. We appreciate your efforts in reviewing relevant literature to understand the context of our work. We remark that, for this work, our main contribution (as you mentioned in strengths) is to provide a unified novel theoretical framework (A-PARL) to address the issues of the existing literature. To support that, we did a rigorous theoritical analysis of the policy alignment problem, which is missing from existing literature (Deep RLHF (Christiano et. al 2017), Pebble (Lee et. al 2021), Surf (Park et. al 2022)) etc. Hence, we performed experiments on the three domains as a proof of concept of our algorithm and theoritical analysis.
>
>
>
> > Weakness 2 :  Could you comment on the generality of the proposed framework? In particular, I am interested in whether this framework can be extended to handle the alignment of language model because the reward model quality is the bottleneck in many cases and most of the existing approaches use only a fixed reward model.
>
>
> **Response to Weakness 2:** This is an excellent point. Indeed, the proposed framework holds promise for extension to the alignment of large language models (LLMs). A significant benefit in the context of LLMs is the deterministic nature of transition dynamics at the lower level. This occurs because we append the new token with the input segment (current state) to generate the next token. This deterministic aspect simplifies and strengthens the framework's application in LLM alignment. We acknowledge this as a valuable direction for future research.

---

> ### Comment · Reviewer_fXDT · 2023-11-14
> **Thanks for the response**
>
> I have read the responses and have no further question. I keep my score toward acceptance.

---

> > ### Author Response · Authors · 2023-11-17
> > **Thank you so much for recommending Acceptance.**
> >
> > Dear Reviewer,
> >
> > We are thankful for your feedback and recommending acceptance of our work.
> >
> > Regards,
> >
> > Authors

---

> > > ### Comment · Reviewer_fXDT · 2023-11-23
> > > **further response**
> > >
> > > Since the ddl is approaching, I just read the review and also discussions from other reviewers. I believe that catching a (statistical) gap in the previous literature is important and interesting. The proposed new bi-level framework for mitigating this gap of statistical dependency is intuitive and is with rigorous theoretical analysis.
> > >
> > > I was not sure about the quality of empirical result so I go through the cited papers and realized that most of the experimental settings are standard in the area. Meanwhile, after reading the discussions with other reviewer, I noticed that the authors have largely enrich the empirical results. I also encourage the authors to further include some missing experiments in the CR version, as suggested by other reviewers.
> > >
> > > Based on the above reasons, I further raise my score to 8.

---

> > > > ### Author Response · Authors · 2023-11-23
> > > >
> > > > Dear Reviewer,
> > > >
> > > > Thank you for your insightful review and for recognizing the significance of our work. Your thorough analysis and constructive suggestions are invaluable. We are motivated by your raised score and committed to incorporating the recommended experiments in our final version.  Thank you once again for your time and efforts in reviewing our paper.
> > > >
> > > > Regards,
> > > >
> > > > Authors

---

### Official Review · Reviewer_iimX · 2023-11-04

**Soundness:** 3 good
**Presentation:** 3 good
**Contribution:** 3 good
**Rating:** 6
**Confidence:** 3

**Summary:**

The paper under review studied policy alignment problem of reinforcement learning.
The authors proposed a bilevel framework to improve the efficiency of the alignment and to better evaluate the alignment.
An algorithm was derived analytically to optimize this bilevel problem, and convergence to stable point was proved.
In addition, simulations was included to demonstrate the capacity of the proposed framework.

**Strengths:**

The bilevel formulation of policy alignment problem is novel and interesting.
The paper is mostly well written, easy to follow (see questions below).
The theoretical derivation is novel and thorough.

**Weaknesses:**

1. The objective of the paper is somewhat confusing to me.
Would the authors please clarify the cause of the misalignment that is considered? and what role does the human agent play?

* Is there a fixed RL problem, but the agent can only get some feedback from a human who partially knows the real reward, and such
feedback changes over time as human learns more about the reward?

* or a dynamically changing RL problem is consider i.e. reward (value function) changes over time,
and one possible cause of such change is due to how the reward is generated (feedback produced by a human's preference).

More precisely, the authors used $\nu$ to parameterize the upper level (evaluating the lower level policy).
What is does each $\nu$ represent?


2. My other concern is experiments are not detailed. Descriptions and simulation set up are all missing. For instance, figure 2,
what precisely do 'walker' or 'door open' or 'button press' aim to do?
how does the misalignment occur? what does oracle represent? what are the upper level objective functions $G(\nu, \theta^*(\nu))$ for these tasks?

**Questions:**

1. On top of page 4, the author mentioned $\nu \in \mathbb{R}^n$. What does $n$ represent here?

2. In the paper, the authors used $H_l$ to represent the horizon of the lower-level trajectory, and $H_u$ to represent the horizon of the lower-level trajectory. Why there are two different horizons? are trajectories used for upper level resampled?

3. On page 7, in assumption 4, what do $\mu, \hat{l}, \hat{\mu}$ represent?

4. In Lemma 1, does eq 21 hold for any choice of convex function $f$ or a specific $f$? also $L_{\pi}$ should be $L_2$?

5. Suggestion: through Lemma 2 and Thm 1, there are too many constant symbols such as $L_1, L_5, L_6$. It might be a good idea to combine some of them together in the main text to make the statement cleaner.

---

> ### Author Response · Authors · 2023-11-14
> **Response to Reviewer iimX [Part-I]**
>
> **General Response:** Thank you for your insightful review. We appreciate your recognition of the novel bilevel formulation in addressing the policy alignment problem in reinforcement learning. Your acknowledgment of the clarity of the paper and the thoroughness of our theoretical derivation is greatly encouraging.
>
> ### Weaknesses:
>
> > **Weakness 1.1:** The objective of the paper is somewhat confusing to me. Would the authors please clarify the cause of the misalignment that is considered? and what role does the human agent play?
>
> **Response to Weakness 1** Thanks for this opportunity to clarify. The misalignment primarily arises in RL due to the misspecified reward function because true reward is usually unknown in the majority of the practical scenarios. A human feedback helps to learn the true reward function as also highlighted in (Christiano et. al 2017, Lee et. al 2021, Park et. al 2022). However, interestingly, the existing approaches of learning from human feedback such as (Deep RLHF, PEBBLE, SURF) suffer from misalignment for not considering the interdependence between the reward and policy learning leading to sub-optimality (as highlighted in our paper, Sec 3.1, 3.2, Figure 1). Incorporating this dependence through a bilevel formulation helps in mitigating the performance gap of prior algorithms, as shown in Figure 2.
>
> Hope this clears the confusion. Please let us know if there are any further confusions regarding this point, we are happy to discuss and elaborate further.
>
>
> >**Weakness 1.2:**  Is there a fixed RL problem, but the agent can only get some feedback from a human who partially knows the real reward, and such feedback changes over time as human learns more about the reward? or a dynamically changing RL problem is considered i.e. reward (value function) changes over time, and one possible cause of such change is due to how the reward is generated (feedback produced by a human's preference).
>
> **Response to Weakness 2.1:** Yes, our formulation considers a fixed RL problem. The problem deals with the setting (similar setting to Deep RLHF, Pebble, Surf etc.) where there is a ***fixed ground truth reward function*** $r_{\nu^*}$ ($\nu^*$ is the optimal reward parameter unknown to RL agent).  The human feedback corresponds to the optimal reward function which is received by the RL agent as preferences on querying the human with relevant trajectory pairs.
>
> In our work, to learn optimal $\nu^*$ from the human feedback, the existing state-of-the-art methods ignore the dependence on the optimal parameter of the policy $\theta^*$. We highlight this drawback and propose to resolve it via formulating a novel bilevel formulation where we iteratively update the parametrized reward function $\nu_t$ at the outer objective (shown in equation 8, 14) by collecting human feedback on trajectories generated by solving the inner optimal policy (shown in equation 15). This is the exact setting of our formulation and is important to note that the true $\nu^*$ doesn't change over time and is fixed, as also followed in previous baselines.
>
>
> > **Weakness 2.2:** More precisely, the authors used to parameterize the upper level (evaluating the lower level policy). What does each $\nu$ represent?
>
>
> **Response to Weakness 2.2** In our formulation, $\nu$ represents the parameter for the reward model $r_{\nu}$ and the objective is to learn the ground truth reward parameter $\nu^*$.
>
>
> > **Weakness 2.3:** My other concern is experiments are not detailed. Descriptions and simulation set up are all missing. For instance, figure 2, what precisely do 'walker' or 'door open' or 'button press' aim to do? how does the misalignment occur? what does oracle represent? what are the upper level objective functions for these tasks?
>
> **Response to Weakness 2.3** Thank you for raising this concern. Since we have followed the exact similar enviromental configuration and setup used in previous benchmarks, we cited the reference (Christiano et. al, Lee at. al, Park et. al) without specific descriptions. But we agree with the reviewer that providing a detailed description will definitely be helpful for the readers and hence we have updated [in Appendix K](https://openreview.net/pdf?id=ByR3NdDSZB) of our paper with relevant details.

---

> > ### Author Response · Authors · 2023-11-17
> > **Response to Reviewer iimX [Part-II]**
> >
> > ## **Questions**
> >
> > > **Question 1:** On top of page 4, the author mentioned $\nu \in R^n$. What does n represent here?
> >
> > **Response to Question 1:** Here $n$ reprsents the dimensionality of the parameter $\nu$ used to parametrize the reward function.
> >
> > > **Question 2:** In the paper, the authors used $H_l$ to represent the horizon of the lower-level trajectory, and $H_u$ to represent the horizon of the lower-level trajectory. Why there are two different horizons? are trajectories used for upper level resampled?
> >
> > **Response to Question 2:** To be more general (in-terms of the analysis) we used different horizon length for the upper and inner level objectives, however the results hold true when H_u = H_l = H.
> >
> > > Question 3 : On page 7, in assumption 4, what do $\mu, \hat{l}, \hat{\mu}$ represent?
> >
> > **Response to Question 3** Here, $\mu$ represents the PL (Polyak- Lojasiewicz) constant for the value function $V_s(\nu, \theta)$ with $\mu > 0$. Then,  $\hat{l}, \hat{\mu}$ represents the bound on the eigen values of the Hessian of the value function (common for various standard policy parametrizations) which we later discuss in Appendix Section I (page 41).
> >
> > > Question 4 : In Lemma 1, does eq 21 hold for any choice of convex function $f$ or a specific $f$? also $L_{\pi}$ should be $L_2$?
> >
> > **Response to Question 4** Equation 21 holds for the standard choice of convex functions $f$ used in f-divergence [1] and the specific choice of functions results in the type of divergence we want to study (for ex: Total variation, Hellinger, etc.). Thanks for pointing the typo, yes $L_{\pi} = L_2$, updated in the draft.
> >
> > > Suggestion 1 : Suggestion: through Lemma 2 and Thm 1, there are too many constant symbols such as $L_1, L_5, L_6$. It might be a good idea to combine some of them together in the main text to make the statement cleaner.
> >
> > **Response to Suggestion 1** Thanks for the very helpful suggestion and we agree that using too many coefficients can confuse the reader and will combine some of them with unified notation for easier read in the updated draft.
> >
> > Reference :
> >
> > [1]. X. Nguyen, M. J. Wainwright, and M. I. Jordan. On surrogate loss functions and f-divergences. The Annals
> > of Statistics, pages 876–904, 2009.

---

> > > ### Author Response · Authors · 2023-11-17
> > > **Request to Reviewer iimX**
> > >
> > > Dear Reviewer,
> > >
> > > Thank you so much for your time and efforts in reviewing our paper. We have addressed your comments in detail and are happy to discuss more if there are any additional concerns. We are looking forward to your feedback and would greatly appreciate you consider raising the scores.
> > >
> > > Thank you,
> > >
> > > Authors

---

> > > > ### Comment · Reviewer_iimX · 2023-11-18
> > > > **Thank you for the response**
> > > >
> > > > Thank you for the detailed response, it has addressed my concerns, score is raised accordingly.

---

> > > > > ### Author Response · Authors · 2023-11-18
> > > > > **Thank you for raising the scores.**
> > > > >
> > > > > Dear Reviewer,
> > > > >
> > > > > Thank you so much for your feedback and for raising the scores toward acceptance.
> > > > >
> > > > > Regards,
> > > > >
> > > > > Authors

---

### Official Review · Reviewer_nxvi · 2023-11-07

**Soundness:** 4 excellent
**Presentation:** 3 good
**Contribution:** 3 good
**Rating:** 6
**Confidence:** 3

**Summary:**

This paper formulates the policy alignment problem in RL (PARL) as a bi-level optimization, where the upper-level optimization corresponds to reward design while the lower-level optimization corresponds to the policy optimization for the designed reward function. Solving the bilevel optimization is challenging, due to the complicatedness of computing the gradient for the upper-level optimization, requiring information about the optimal policy for the given reward function. A practical algorithm is presented with approximations that use K-step policy gradient updates, and sample-based estimation of Jacobian/Hessian. Convergence analysis is provided, achieving the rate of O(1/T). In the experiments, the proposed A-PARL achieves near-oracle performance in a faster time than the baselines.

**Strengths:**

1. This paper introduces a mathematical framework for policy alignment in RL as a bilevel optimization, which includes useful applications like RLHF.
2. A theoretical convergence guarantee is provided for the proposed method.
3. In the experiments, it outperforms the baselines, showing better sample efficiency.

**Weaknesses:**

1. Empirical evaluations could have been more thorough. In Figure 2, only three domains were considered (2 manipulation / 1 locomotion), which is much smaller than the ones in the baseline papers. For example, in the PEBBLE/SURF papers, they conducted experiments in 9 domains (6 manipulation / 3 locomotion). I am curious if A-PARL still outperforms the baselines in other domains.
2. Some part of the algorithm is unclear. For example, it seems that gradient/Jacobian/Hessian estimation (Eq 15-17) requires 'on-policy' samples for their computations. Still, I was first assuming the lower-level policy optimization is done using an off-policy RL algorithm, as in the previous work (e.g. PEBBLE). Then, is my understanding correct that upper-level optimization is done using on-policy samples while the lower-level optimization is done in an off-policy way? Does lower-level optimization also use on-policy updates (Algorithm 1 line 5 seems to perform on-policy updates)? If so, I am curious how the on-policy algorithm could have shown better sample efficiency than the off-policy baselines.
3. Lack of ablation studies. For example, to see the efficiency of A-PARL more properly, it would be great to see the results using varying the number of feedbacks. In the current form of the paper, it is even unclear how many feedbacks were used or how frequently the feedback was collected in the experiments.
4. Another concern is the scalability of the algorithm. Given that the method requires Hessians, it seems the complexity of the method is quadratic w.r.t. the number of parameters of the policy network. The method may not be applicable to the setting where the network size is large (e.g. millions of parameters).

**Questions:**

1. Please see the weaknesses section above.
2. Though it was interesting to see the corrected gradient updates for the upper-level optimization, it was hard to interpret the terms directly. Could you provide some intuitive interpretation of the additional terms in the upper-level gradient ($\tilde M^K(\nu_t)$)?
3. Without the $U_\nu(\tau) \sum \tilde M^K(\nu_t) \nabla_\theta f_h(\theta^K(\nu_t))$ term in Eq (14), does A-PARL reduce to the existing method or not?

---

> ### Author Response · Authors · 2023-11-14
> **Response to Reviewer nxvi [Part-I]**
>
> **General Response:** Thank you for your insightful review. Your detailed summary accurately captures the essence of our approach in formulating the PARL as a bi-level optimization problem. We are grateful for your recognition of the practicality and theoretical rigor of our algorithm, as well as its performance in the experiments. Your constructive comments are highly valuable, and we address them in detail as follows.
>
> ### Weaknesses:
>
> > **Weakness 1:** Empirical evaluations could have been more thorough. In Figure 2, only three domains were considered (2 manipulation / 1 locomotion), which is much smaller than the ones in the baseline papers. For example, in the PEBBLE/SURF papers, they conducted experiments in 9 domains (6 manipulation / 3 locomotion). I am curious if A-PARL still outperforms the baselines in other domains.
>
> **Response to Weakness 1** : We agree that it is possible to conduct more experiments on the remaining baselines. However, for this work, our primary focus is to provide a unified novel theoretical framework (A-PARL) and a rigorous theoritical analysis of the policy alignment problem, which is missing from existing literature (RLHF (Christiano et. al 2017), Pebble (Lee et. al 2021), SURF (Park et. al 2022) etc.). Hence, we performed experiments on three domains (with different level of hardness) as proof of concept of our algorithm and theoretical analysis.
>
> ***New Experiment Results*:** However, we remark that our performance is transferable to other environments as well. As mentioned by the reviewers, we added additional hard environments like Cheetah (DMC) and compared the performance of A-PARL with baselines [in Figure 6 in Appendix K (marked in blue)](https://openreview.net/pdf?id=ByR3NdDSZB). We observe that A-PARL outperforms existing baselines in both environments.
>
>
>
> **Note**: Due to time constraints, we have only added Cheetah (Locomotion, DMSuite) for now, but we are running further experiments on more hard environments (like visual-Walker, Sweep Into) and will try our best to add them in the final revised draft.
>
> > **Weakness 2:** Some part of the algorithm is unclear. For example, it seems that gradient/Jacobian/Hessian estimation (Eq 15-17) requires 'on-policy' samples for their computations. Still, I was first assuming the lower-level policy optimization is done using an off-policy RL algorithm, as in the previous work (e.g. PEBBLE). Then, is my understanding correct that upper-level optimization is done using on-policy samples while the lower-level optimization is done in an off-policy way? Does lower-level optimization also use on-policy updates (Algorithm 1 line 5 seems to perform on-policy updates)? If so, I am curious how the on-policy algorithm could have shown better sample efficiency than the off-policy baselines.
>
> **Response to Weakness 2** This is a good point. We note that in our bilevel formulation,  the inner level is the standard RL policy optimization problem and can be solved using any standard RL algorithm either on-policy or off-policy. For the sake of analysis and establishing a connection to the most basic setting (as in RLHF (Christiano et. al 2021)), our policy learning update in Algorithm 1 is an on-policy. For the experiments and to have fair comparisons with Pebble and Surf, we use the same backbone (SAC which is off policy) with similar configurations and hyperparameters so that the results are comparable.
>
> We remark that irrespective of the algorithm we use to solve the inner-level RL problem, a key innovation in our approach lies in incorporating the gradient $\nabla_{\nu} \theta^*(\nu)$ (equation 12) in the outer objective (equation 8, 14), which helps to learn the true optimal reward and close the gap in the existing literature.
>
> > Weakness 3 : Lack of ablation studies. For example, to see the efficiency of A-PARL more properly, it would be great to see the results using varying the number of feedbacks. In the current form of the paper, it is even unclear how many feedbacks were used or how frequently the feedback was collected in the experiments.
>
> **Response to Weakness 3:** In the current setting, we compare with the exact number of default feedbacks: *Walker (max_feedback = $100$), Door Open (max_feedback = $1000$) and Button Press (max_feedback = $20000$) as typically used in (RLHF, Pebble, SURF etc.)* for the three environments mentioned.
>
> ***New Ablation Study:*** As requested by the reviewer, we did additional experiments by varying the number of feedbacks as shown [in  Figure 7 in Appendix K.1 in the updated draft](https://openreview.net/pdf?id=ByR3NdDSZB). It shows that more human feedback results in better performance as expected. We will add a much more detailed version of this ablation study with respect to human feedback in the final version of the paper.

---

> > ### Author Response · Authors · 2023-11-14
> > **Response to Reviewer nxvi [Part-II]**
> >
> > > Weakness 4 : Another concern is the scalability of the algorithm. Given that the method requires Hessians, it seems the complexity of the method is quadratic w.r.t. the number of parameters of the policy network. The method may not be applicable to the setting where the network size is large (e.g. millions of parameters).
> >
> > **Response to Weakness 4** : This is a very good point, thanks for raising this concern. We agree that our algorithm requires second-order estimates and would suffer in very large scale settings. But since our formulation is a first step towards rigorously formulating the policy alignment problem, we stick to the most fundamental and basic algorithmic setting to emphasize on the formulation and unified algorithm. However, we want to highlight that with the recent developments of scalable libraries such as Torchopt (Hypergrad), it is comparatively easier to estimate these terms approximately. Furthermore, scaling our approach to large scale is a valid scope of future research on which we are currently working on as a follow up to this work.
> >
> >
> > ## **Questions:**
> >
> >
> > > Question 1: Though it was interesting to see the corrected gradient updates for the upper-level optimization, it was hard to interpret the terms directly. Could you provide some intuitive interpretation of the additional terms in the upper-level gradient $\tilde{M^K}(\nu_{t})$
> >
> > **Response to Question 1** : Thank you for raising this comment. To understand the inutition behind the additional term in the upper-level gradient $U_{\nu}(\tau)\cdot \sum_{t =0}^{H_u-1} [\widetilde{M}^K(\nu_t) \nabla_{\theta} f_h(\theta^K(\nu_t))]$ arising in Equation 8, 14, let's take a step back to the term $\nabla_{\nu} \log \pi_{\theta^*(\nu)}(a_h|s_h)$ in equation 12 which eventually results in this additional term $\tilde{M^K}(\nu_{t})$ (implicit gradient).
> >
> > Note that the term $\nabla_{\nu} \log \pi_{\theta^*(\nu)}(a_h|s_h)$ is extremely important and gives a direct connection between the reward parameter $\nu$ and the optimal policy parameter $\theta^*(\nu)$. Intuitively, it provides information about the performance of the optimal agent policy under the specified reward. Utilization of this connection was missing from the existing literature. Thus, with this additional term, we perform a new proposed update in the outer objective which respects the inner-agent's performance under that specified reward in-addition to maximizing the likelihood of human preference (for RLHF application), which is the unique aspect of our formulation.
> >
> >
> > > Question 2: Without the $\theta \in U_{l}(\tau) \Rightarrow \sum_{\nu_{t}}^K \tilde{M^K}(\nu_{t}) \nabla_{\theta} f_h(\theta^K (\nu_{t}))$ term in Eq (14), does A-PARL reduce to the existing method or not?
> >
> > **Response to Question2** : You are correct, if we ignore the first term $U_{\nu}(\tau)\cdot \sum_{t =0}^{H_u-1} [\widetilde{M}^K(\nu_t) \nabla_{\theta} f_h(\theta^K(\nu_t))]$ inside the expectation in equation 14, A-PARL will reduce to standard RLHF frameworks (Christiano et. al 2017, Lee et. al 2021, Park et. al 2022 etc.) and thus our formulation generalizes the existing approaches and captures a unique missing aspect of existing formulations in the literature.

---

> > > ### Author Response · Authors · 2023-11-17
> > > **Request to Reviewer nxvi**
> > >
> > > Dear Reviewer,
> > >
> > > Thank you so much for your time and efforts in reviewing our paper. We have addressed your comments in detail and are happy to discuss more if there are any additional concerns. We are looking forward to your feedback and would greatly appreciate you consider raising the scores.
> > >
> > > Thank you,
> > >
> > > Authors

---

> > > > ### Comment · Reviewer_nxvi · 2023-11-23
> > > > **Thanks for the response**
> > > >
> > > > Thanks for your response, which addressed most of my concerns. I would like to raise my score accordingly.
> > > > However, one thing is still not clear to me: Based on Table 2 in the Appendix, it uses 1024 hidden size (I assume the number of hidden layers is at least 2), which means that there are at least 1M weight parameters (1024 x 1024 $\approx$ 1M). Even with millions of parameters, how was the Hessian computed (Eq 16) in the experiments? Furthermore, the matrix inversion should be computed for the estimated Hessian (Eq 14), which requires O(N^3) complexity. This would be practically intractable when N is a million. Could you clarify how your method was implemented to address this issue?

---

> > > > > ### Author Response · Authors · 2023-11-23
> > > > >
> > > > > **Response:** Thank you so much for your feedback and raising the scores. Regarding the Hessian inverse computation, we agree that it is a valid point to consider, especially in large-scale settings. It is arising due to the need to perform implicit differentiation for the outer objective, fortunately, in the bilevel optimization and meta-learning literature, this problem has been widely considered (Chen et al.,2022; Kwon et al., 2023; Li et al., 2022b; Ji et al., 2021; Cao et al., 2023; Akhtar et al., 2022; Ghadimi & Wang, 2018b). A Standard repository [Torchopt](https://torchopt.readthedocs.io/en/latest/) (or [BOML](https://github.com/dut-media-lab/BOML) in Tensorflow) is widely used to evaluate the meta gradient (which estimates Hessian inverse via approximations like Conjugate gradient, Gradient unrolling etc.), which we leverage for our implementations. We will add the specific implementation details with references (packages used) and links in our updated draft.
> > > > >
> > > > > We want to thank the reviewer once again for the insightful questions that helped in improving the quality of our draft.
> > > > >
> > > > > Regards,
> > > > >
> > > > > Authors

---

### Official Review · Reviewer_fCoY · 2023-11-10

**Soundness:** 3 good
**Presentation:** 2 fair
**Contribution:** 3 good
**Rating:** 8
**Confidence:** 3

**Summary:**

[This review is submitted upon an emergency review request.]

The paper introduces PARL (Policy Alignment Reinforcement Learning), a bilevel optimization framework for addressing policy alignment in reinforcement learning. It identifies and aims to fill a gap in existing methods by considering the relationship between policy alignment and data generated from policy trajectories. PARL comprises an upper level focused on reward design and a lower level dedicated to policy optimization. The paper also presents a new algorithm, A-PARL, and reports improved sample efficiency in tests conducted in environments like the DeepMind control suite and MetaWorld.

**Strengths:**

- The paper's approach to formulating preference-based reinforcement learning as a bilevel optimization problem is intuitive and appears well-reasoned. This framework effectively captures the influence of the lower-level policy optimization on the upper-level objective.

- There is a commendable theoretical analysis of the framework and the proposed A-PARL algorithm, which contributes to the understanding of the methodology and its potential impacts.

**Weaknesses:**

The paper lacks empirical evidence to substantiate the claim that their method effectively handles the shift in policy distribution at the lower level. While it shows improved final policy return performances, more diagnostic evaluations are needed to conclusively attribute these improvements to the framework's ability to manage lower-level distribution shifts and alignment issues.

**Questions:**

- What values of K and N were used in Algorithm 1 for the experiments?
- Could you elaborate on how these parameters were tuned and the impact of selecting different values?
- Specifically, how does a larger K value, which implies more accurate solving of the lower problem but at the cost of additional environment trajectories, affect the overall outcomes? What trade-offs are observed with varying K values?

---

> ### Author Response · Authors · 2023-11-14
> **Response to Reviewer fCoY**
>
> **General Response:** We sincerely appreciate the reviewer's thoughtful feedback on our paper. We would specfically like to acknowledge the reviewer for identifying the major motivation and contribution of our work, which is to characterize the dependence of the policy optimization on the upper-level objective and provide a detailed theoritical analysis of the proposed framework. We are pleased that the reviewer recognized the significance and relevance of this contribution to the field of policy alignment.
>
>
> ### Questions:
>
> > **Questions:** What values of K and N were used in Algorithm 1 for the experiments?
> Could you elaborate on how these parameters were tuned and the impact of selecting different values?
> Specifically, how does a larger K value, which implies more accurate solving of the lower problem but at the cost of additional environment trajectories, affect the overall outcomes? What trade-offs are observed with varying K values?
>
> **Responses:**  Thank you for this comment. For the experiments we choose the default settings as used in (RLHF (Christiano et. al 2017), Pebble (Lee et. al 2021), SURF (Park et. al 2022) etc.). The exact values are for Walker (N = $100$), Door Open (N = $1000$) and Button Press (N = $20,000$)  for the human feedback. As requested by some other reviewers, we have also varied the feedback count and performed additional experiments, results are updated [in Appendix K (marked in blue)](https://openreview.net/pdf?id=ByR3NdDSZB).
>
> Regarding the value of $K$, as you correctly pointed out,  it represents the number of gradient updates for the inner policy optimization loop i.e., how much we optimize the policy for the given reward. Interestingly, we are able to derive a precise value of $K = t$ ($t$ : outer iteration count) from our analysis (equations 58, 59 section F in Appendix) and use the same in experiments. As pointed out by the reviewer, it's also very intuitive and represents that as the reward becomes more and more closer towards the optimal, we increase the value of $K$ i.e more accurately solve the inner problem. We remark that selecting any other value of $K$ would result in a tradeoff in terms of optimality of the inner level problem.
>
>
> Thanks for the insightful questions and we will add the discussion to our revised draft.

---

### Meta-Review · Area_Chair_yJyM · 2023-12-09

**Metareview:**

Paper introduces a novel bilevel optimization framework for addressing policy alignment in reinforcement learning. It fills a gap in existing methods with a new algorithm, A-PARL, with improved sample efficiency in benchmark environments. Paper is well-written with detailed explanations, and also demonstrate empirical efficacy of the proposed method

**Justification For Why Not Higher Score:**

Lacking further ablation studies and some references are missing in the literature review section. Bilevel optimization problem may seem non-trivial to implement in scalable problems.

**Justification For Why Not Lower Score:**

Paper has novelty both on algorithms and demonstrated improved sample efficiency both algorithmically and empirically. Sufficient contributions to be accepted as a conference paper.

---

### Decision · Program_Chairs · 2024-01-16

Accept (poster)